# Constriction imposed by basement membrane regulates developmental cell migration

Ester Molina López[1], Anna Kabanova[1,2], Alexander Winkel[3], Kristian Franze[3,4,5], Isabel M. Palacios[6], María D. Martín-Bermudo [1]*

1 Centro Andaluz de Biología del Desarrollo CSIC-University Pablo de Olavide, Sevilla, Spain, 2 Department Physiology of Cognitive Processes, MPI for Biological Cybernetics, Tübingen, Germany, 3 Department of Physiology, Development and Neuroscience, University of Cambridge, Cambridge, United Kingdom, 4 Institute of Medical Physics and Micro-Tissue Engineering, Friedrich-Alexander Universität Erlangen-Nürnberg, Erlangen, Germany, 5 Max-Planck-Zentrum für Physik und Medizin, Erlangen, Germany, 6 School of Biological and Behavioural Sciences, Queen Mary University of London, London, United Kingdom

* mdmarber@upo.es

**Data Availability Statement:** All relevant data are within the paper and its Supporting Information files.

## Abstract

The basement membrane (BM) is a specialized extracellular matrix (ECM), which underlies or encases developing tissues. Mechanical properties of encasing BMs have been shown to profoundly influence the shaping of associated tissues. Here, we use the migration of the border cells (BCs) of the *Drosophila* egg chamber to unravel a new role of encasing BMs in cell migration. BCs move between a group of cells, the nurse cells (NCs), that are enclosed by a monolayer of follicle cells (FCs), which is, in turn, surrounded by a BM, the follicle BM. We show that increasing or reducing the stiffness of the follicle BM, by altering laminins or type IV collagen levels, conversely affects BC migration speed and alters migration mode and dynamics. Follicle BM stiffness also controls pairwise NC and FC cortical tension. We propose that constraints imposed by the follicle BM influence NC and FC cortical tension, which, in turn, regulate BC migration. Encasing BMs emerge as key players in the regulation of collective cell migration during morphogenesis.

## Introduction

The basement membrane (BM) is a dense, sheet-like form of an extracellular matrix (ECM) that underlies the basal side of epithelial and endothelial tissues and enwraps most developing organs [1,2]. BMs are highly conserved across evolution and are composed of a core set of proteins, including the secreted glycoproteins laminin, type IV collagen (Coll IV), entactin/nidogen, and the heparan sulfate proteoglycan Perlecan [3]. Originally viewed as a static support ECM, BMs have recently been proven to act as active regulators of tissue shape and homeostasis.

Encasing BMs regulate the morphogenesis of the tissues they enwrap both molecularly, by modulating signaling to cells through surface receptors, and mechanically, by imposing

**Funding:** This work was funded by the Spanish Minister of Science and Innovation (MCIN), grant numbers PID2019-109013GB-100 and BFU2016-80797R, to MDMB and by the regional Agency Fundación Pública Progreso y Salud (Junta de Andalucía), grant number P20_00888, to MDMB. EML received a salary from MCIN (BFU2016-80797R). The funders had no role in study design, data collection and analysis, decision to publish, or preparation of the manuscript.

**Competing interests:** "The authors have declared that no competing interests exist."

**Abbreviations:** AP, anterior-posterior; BC, border cell; BM, basement membrane; ECM, extracellular matrix; FC, follicle cell; FE, follicular epithelium; GFP, green fluorescent protein; NC, nurse cell; ROI, region of interest; StFC, stretched FC.

patterned constricting forces. The best studied developmental function attributed to forces exerted by enwrapping BMs is the regulation of cell and organ shape (reviewed in [4]). Mechanically, BMs mediate organ shape by transmitting tension through interconnected cells. Involvement of BMs in organ shaping is well exemplified in the development of the *Drosophila* central nervous system, wing, and egg chamber [5–9]. Changes in the levels of several BM components, such as Col IV or Laminins, affect the shape of these tissues. Similarly, during mouse mammary gland duct development, BM accumulates at the base of duct buds, thereby constricting and elongating them (reviewed in [10]). Finally, BMs also play critical roles in shaping biological tubes, including *Drosophila* embryonic salivary glands and renal tubes [11,12] and the *Caenorhabditis elegans* excretory system [13]. Besides changing in shape, cells of developing organs encased by BMs also undergo migratory processes. This is the case of the distal tip cell of the *C. elegans* somatic gonad (reviewed in [14]), the avian cranial neural crest cells [15], or some cell populations during vertebrate branching morphogenesis (reviewed in [16]). However, in contrast to the wealth of information on the role of BMs in cell migration when acting as a substratum, little is known about their function in cell movement when acting as a "corset" encasing developing tissues.

The egg chamber (or follicle) of the *Drosophila* ovary, the structure that will give rise to an egg, has proven to be an excellent model system for investigating the in vivo roles of the forces exerted by encasing BMs in organogenesis (reviewed in [17]). Egg chambers are multicellular structures that consist of 16-germline cell cysts—15 nurse cells (NCs) and 1 oocyte—surrounded by a monolayer of somatic follicle cells (FCs), termed the follicular epithelium (FE) (Fig 1A and 1B). Each egg chamber progresses through 14 developmental stages (S1–S14) [18]. The apical side of FCs faces the germline, while the basal side contacts a BM produced by the FCs themselves, the follicle BM. The follicle BM of the *Drosophila* ovary contains Laminins, Coll IV, Perlecan, and Nidogen [19–22]. Initially, the egg chamber is spherical but from S5 it elongates along its anterior-posterior (AP) axis, to acquire the final elliptical shape of the egg. During elongation, FCs move collectively over the inner surface of the follicle BM, a process that has been termed "global tissue rotation" [7]. Rotation is accompanied by alignment of basal actin bundles and polarized secretion of BM components, which become organized in circumferential fibrils oriented perpendicular to the follicle's AP axis [7,23]. The circumferential arrangement of fibrils is thought to act as a "molecular corset" that constrains follicle growth in the direction of the alignment, thus driving egg chamber elongation. Around S9, once global tissue rotation has stopped and the circumferential fibrils of the corset are fully polarized and formed, a group of 6 to 10 anterior FCs, the border cells (BCs), delaminate from the epithelium and perform another type of directed collective cell migration ([18,24], Fig 1B and 1C). BCs extend actin-rich protrusions and move between the NCs until they reach the anterior membrane of the oocyte, following a central path. Although chemical cues produced by the oocyte were thought to be sufficient to direct BC migration, 3D reconstruction of egg chambers have recently shown that microtopography is also important [25]. Thus, while chemo-attractants primarily guide posterior movement, tissue architecture, and more specifically multicellular junctures, which present a lower energy barrier for movement, steer them centrally [25]. During BC migration, NCs exert compressive forces over the BCs, influencing their migration [26]. Additionally, enzymatic removal of the follicle BM, using collagenase, has been shown to affect NC shape [27]. In this context, one could speculate that forces exerted by the follicle BM over the NCs could in principle also influence BC migration. However, this has never been tested.

Here, we show that increasing or reducing the stiffness of the follicle BM, by altering the levels of the BM components laminins or Coll IV, inversely affect BC migration speed. In addition, we found that follicle BM stiffness affects protrusion dynamics and migration mode.

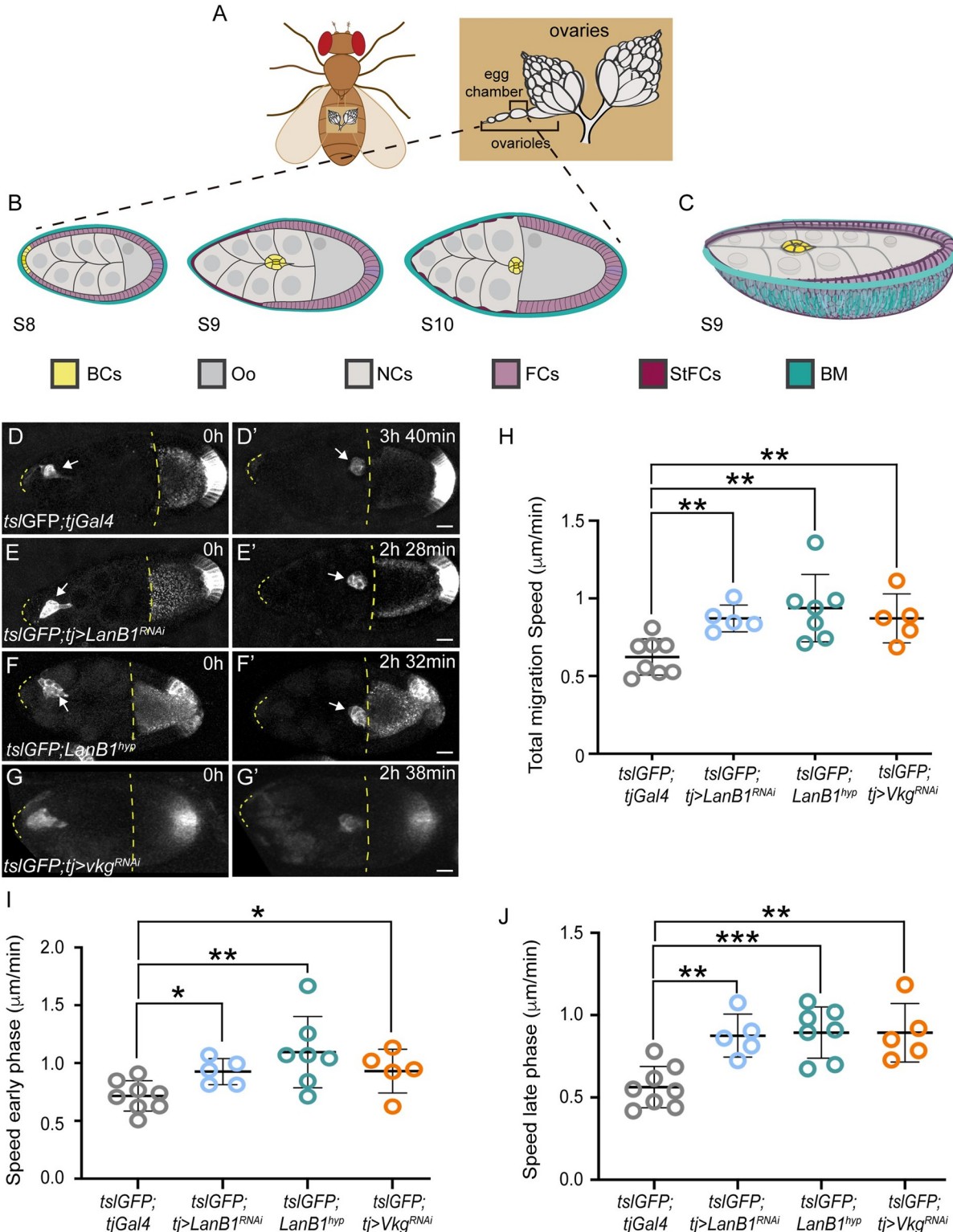

**Fig 1. Reducing Laminins or Col IV levels results in increased BC migration speed.** (A) Position of ovaries within a *Drosophila* female (left) and schematic of *Drosophila* ovaries showing an ovariole and its egg chambers (right). BC, border cells; Oo, oocyte; NCs, nurse cells; FCs, follicle cells; StFCs, stretch FCs; BM, basement membrane. (B) Diagram of an ovariole showing stages 8, 9, and 10 egg chambers (S8–S10). Anterior is left, posterior right. (C) Three-dimensional diagram of a S9 egg chamber showing the migration of the BCs between the NCs. (D–G') Stills taken from live imaging of migrating BCs from egg chambers of the indicated genotypes. Discontinuous yellow lines

demarcate the region between the anterior border of the egg chamber and the oocyte anterior membrane. (H–J) Quantification of BC migration speed during the whole migratory process (H) and during the early (I) and late (J) phases in egg chambers of the indicated genotypes. The statistical significance of differences was assessed with a *t* test, ** *P* value < 0.01. Horizontal and vertical lines indicate mean and SD, respectively. Scale bars in A', B', C' and D', 20 μm. The raw data underlying panels H–J are available in S1 Data.

Furthermore, our results show that follicle BM stiffness controls pairwise FC and NC cortical tension. Finally, we show that a direct manipulation of FC tension affects inversely BC migration speed. Based on these results, we propose that constraints imposed by the follicle BM can affect FCs and NCs cortical tension, which, in turn, influences BC migration. Our results unravel a new role for the mechanical properties of the BMs enclosing developing organs, the regulation of the intercellular forces controlling collective cell migration. In addition, this work provides new insights towards our understanding of the remarkable and assorted ways whereby BMs regulate cell migration during morphogenesis.

## Results

### The stiffness of the follicle BM influences BC migration

To test the role of the follicle BM on BC migration, we reduced the levels of 2 of its main components, Laminins or Coll IV. Laminins are large heterotrimeric glycoproteins that contain 1 copy each of an α, β, and γ chain. *Drosophila* contains 2 different Laminin trimers, each composed of one of the two α chains, encoded by the *wing blister* (*wb*) and *Laminin A* (*LanA*) genes, and the shared β and γ chains, encoded by the *Laminin B1* (*LanB1*) and *Laminin B2* (*LanB2*) genes, respectively [28–31]. To reduce Laminin levels in follicles, we either expressed a LanB1 RNAi in all FCs by means of the *traffic jam* (*tj*) Gal4 line (*tj>LanB1*$^{RNAi}$) or used a hypomorphic viable condition for the locus, *LanB1*$^{28a}$/*l(2)K05404* (hereafter referred to as *LanB1*$^{hyp}$, Table 1), as both have been shown to drastically reduce LanB1 levels up to S8 [11,32]. We have previously shown by atomic force microscopy measurements that follicle BM stiffness was significantly diminished in these mutant conditions, from the germarium up to S8 [32]. As here we found that LanB1 levels in the mutant follicles were kept constant and significantly inferior to controls until the end of oogenesis (S1 Fig), we expected that the reduction in BM stiffness would sustain up to S9, when BCs are migrating. To confirm this, we utilized atomic force microscopy and measured the apparent elastic modulus, *K*, as a proxy for BM stiffness [32]. Force-distance curves were analyzed for an indentation depth of 200 nm, because egg chambers are composite materials with several layers, and structures away from the surface farther than approximately 1/10 of the indentation depth do not contribute to *K* [33]. As expected, we found that the *K* values of laminin-depleted follicles (*n* = 16 from 7 different egg chambers) were significantly smaller compared with controls (*n* = 12 from 6 different egg chambers, S2A and S2B Fig).

Another key component of BMs is type IV collagen, a unique member of the collagen superfamily, which is composed of different a1 chains that form distinct heterotrimers. The *Drosophila* Coll type IV trimer consists of two α1 chains and one α2 chain, encoded by the genes *Collagen at 25C* (*Cg25C*) and *Viking* (*Vkg*), respectively [34–36]. To decrease Coll IV levels, we expressed a Vkg RNAi in all FCs using the *tj*Gal4 driver (*tj>Coll IV*$^{RNAi}$, Table 1), an approach previously shown to efficiently reduce both Coll IV levels and BM stiffness up to S8 [7,37]. In addition, as it was the case for Laminins, we found that the *K* values of *tj>Coll IV*$^{RNAi}$ follicles were kept significantly smaller than those of controls at S9 (S2A and S2B Fig).

These results demonstrated that the approach here used to decrease Laminins and Col IV levels in follicles caused a reduction in BM stiffness at S9, which is when BCs are migrating. This allowed us to test the effects of lessening BM stiffness in BC migration.

**Table 1. Key resources table.**

| Reagent or resource | Source | Identifier |
| --- | --- | --- |
| Antibodies | | |
| Rabbit anti-pSqh | Cell Signaling | 3671S |
| Anti-Rabbit Cy3 | Jackson ImmunoReseach Laboratories | 211-502-171 |
| Anti-Rabbit Cy2 | Jackson ImmunoReseach Laboratories | 711225152 |
| Anti-LanB1 | Abcam | EPR3189 |
| Chemicals, peptides, and recombinant proteins | | |
| Rhodamine-Phalloidin | Molecular Probes | R37112 |
| Hoechst | Thermo Fisher | H21491 |
| Experimental models: Organisms/strains | | |
| $LanB1^{28a}$ (II) | [11] | N/A |
| l(2)K05404 | Bloomington Drosophila Stock Center | BDSC 13957 |
| $Kelch^{ED1}$ | Bloomington Drosophila Stock Center | 4893 |
| $dic^1$ | Bloomington Drosophila Stock Center | 4223 |
| UAS-LanB1 RNAi | VDRC | V23119 |
| UAS-Vkg RNAi | Bloomington Drosophila Stock Center | BDSC 50895 |
| UAS-Abi RNAi | DGRC Kyoto | 9749R |
| UAS-EHBP1mCh | Bloomington Drosophila Stock Center | BDSC 67145 |
| mirror Gal4 | Bloomington Drosophila Stock Center | BDSC 63048 |
| slbo Gal4 | Bloomington Drosophila Stock Center | BDSC 8915 |
| C306 Gal4 | Bloomington Drosophila Stock Center | BDSC 3743 |
| Tsl Gal4 | [38] | N/A |
| nos Gal4 | Bloomington Drosophila Stock Center | BDSC 4937 |
| ResilleGFP | Kindly provided by Prof. Wieschaus | N/A |

To quantify BC migration in the different conditions in which the BM stiffness was altered, we performed live imaging of control and Laminin or Coll IV-depleted egg chambers. To visualize BCs, we generated transgenic flies that expressed the green fluorescent protein (GFP) driven by an enhancer of the *torso-like* (*tsl*) gene, which is active in BCs and in a group of posterior FCs (*tsl*GFP, see Materials and methods, [38]). Quantification of BC migration speed from live imaging analysis revealed that BCs from *tsl*GFP; *tj>LanB1*$^{RNAi}$ (*n* = 7), *tsl*GFP; *LanB1*$^{hyp}$ (*n* = 5), or *tsl*GFP; *tj>Vkg*$^{RNAi}$ (*n* = 5) egg chambers moved faster than BCs from control follicles (*tsl*GFP; *tj>Gal4*, *n* = 8) (Fig 1D–1H, S1 Movie, see Materials and methods). In all experimental egg chambers, BCs completed their migration and reached the oocyte as in controls.

BC migration to the oocyte has been divided in 2 phases. In an early phase—from detachment until halfway to the oocyte—movement is more streamlined and sliding. In the late phase—from midway to contacting the oocyte—movement is slower with clusters rotating in place [39,40]. We found that reducing Laminins or Col IV levels increased the migration speed of both phases (Fig 1I and 1J). Since we found that BC migration was similarly increased when reducing either Laminins or Coll IV levels and as Laminins were required for proper Coll IV deposition in the ovary [32], we concentrated our analysis on laminin-depleted egg chambers.

These results suggest the existence of a direct correlation between BM stiffness and BC migration speed. However, the general FC driver *tjGal4* directs expression in all FCs, including BCs. Even though Laminins are not expressed in BCs during their migration [41] (S1A Fig), to completely rule out the possibility that the faster BC migration phenotype observed in *tj>LanB1*$^{RNAi}$ follicles was due to a possible requirement of this BM component in BCs, we

analyzed the effects of expressing a *LanB1* RNAi specifically in BCs. To achieve efficient RNAi knockdown in BCs, we combined the BC *Gal4* line *slboGal4* [24] with 2 other *Gal4* lines, *C306Gal4* and *torso-likeGal4* (*tslGal4*), which are expressed before BCs detach from the FE ([24,38], Table 1). As expected, we could not detect any LanB1 signal in the BCs of these follicles *triple>LanB1^RNAi* (S3A and S3A' Fig). In addition, live imaging of egg chambers coexpressing the *LanB1^RNAi* and a GFP, to visualize the BCs, showed that BCs from the *triple>GFP; LanB1^RNAi* egg chambers (*n* = 6) moved at similar speed than controls (*triple>GFP; n* = 5, S3 Fig and S2 Movie). This result indicated that Laminins do not have an essential function in BCs, a finding that is consistent with the reported absence of Laminins expression in BCs during their migration [41].

To further test the idea that the stiffness of the BM affects BC migration, we analyzed the effect of increasing BM stiffness. To this end, we used the *tjGal4* to overexpress an mCherry tagged version of *EHBP1*(*EHBP1mCh*) in all FCs, an approach known to increase Coll IV fibril deposition and BM stiffness ([8], Table 1). In fact, in agreement with those studies, we found that the *K* values of *tj>EHBP1mCh* follicles obtained by AFM were significantly higher than those of controls (S2B Fig). Live time-lapse imaging showed that the overexpression of *EHBP1mCh* in all FCs delayed BC migration (*tslGFP; tj> EHBP1mCh*, S4A–S4B' Fig, S3 Movie, *n* = 6). However, as mentioned above, the general driver *tjGal4* drives expression in all FCs, including BCs (S4 Fig). In this context, the BC migration defects observed in *tslGFP; tj>EHBP1mCh* follicles could also be due to an increase in Coll IV levels in BCs, rather than just a rise in follicle BM stiffness [37]. Thus, to directly assess the impact of BM stiffness in BC migration, we used the *mirrorGal4* (*mirrGal4*) driver, whose expression is restricted to a central region of the FE [42]. A previous study showed that by S8, the BM at the central region was stiffer than that at terminal regions [37]. Here, we found that this difference persisted into S9 egg chambers (S2D Fig). In addition, we found that this difference was higher when *EHBP1mCh* was expressed in the central region of the follicle using the *mirrGal4* line compared to controls (*mirr>EHBP1mCh*, S2D and S2E Fig). This result demonstrated, in agreement with previous results [37], that overexpressing *EHBP1mCh* locally increased BM stiffness. Next, we analyzed BC migration in these experimental conditions and found that, while BCs migrated normally anterior to the *mirror* region in the *tslGFP; mirr>EHBP1mCh* egg chambers (Fig 2A–2C, *n* = 8), migration speed was reduced once BCs reached the *mirror* region (Fig 2D, *n* = 8, S4 Movie). Furthermore, while migration was always completed in controls follicles (*n* = 6), migration in the *mirror* region was halted in 20% of the experimental cases analyzed (Fig 2E, *n* = 9).

## Effect of the BM stiffness on cellular protrusions dynamics and migration mode

As mentioned in the Introduction, BCs extend actin-rich protrusions while migrating between the NCs. As these protrusions extend between NCs, it is logical to think that any constraint(s) could influence the formation and dynamics of these protrusions. Thus, we next analyzed by live imaging the behavior of BC cellular protrusions in control and experimental clusters throughout the migratory process. In agreement with previous results [39,43], we found that BC protrusions in control egg chambers showed an overall front bias and that the protrusion at the front of the cluster was longer than protrusions from other areas of the cluster (*n* = 7, Fig 3A, 3E, and 3K and S1 Movie). This is maintained when manipulating BM stiffness (Fig 3E and 3F). However, BCs from laminin-depleted egg chambers (*tslGFP; tj>LanB1^RNAi*) showed a significant increase in the number of protrusions extended laterally (*n* = 6, Fig 3B and 3E). Moreover, BCs from *tslGFP; mirr>EHBP1mCh* egg chambers showed a decrease in the

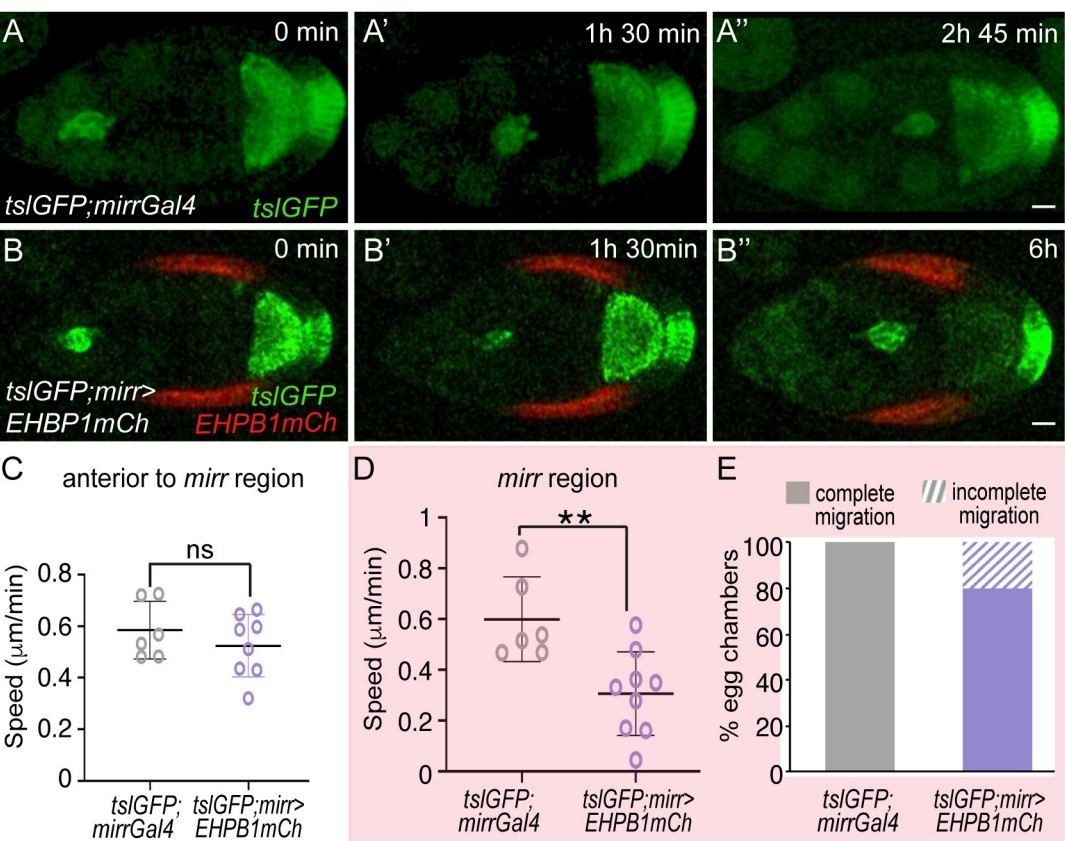

**Fig 2. Increasing EHBP1 levels in FCs results in reduced BC migration.** (A–B") Stills taken from live imaging of migrating BCs from egg chambers of the indicated genotypes. (C, D) Quantification of BC migration speed in egg chambers of the indicated genotypes in the area anterior to (C) and at the (D) *mirr* expressing region. (E) Quantification of egg chambers with complete BC migration. The statistical significance of differences was assessed with a *t* test, * *P* value < 0.05. Horizontal and vertical lines indicate mean and SD, respectively. Scale bars in A" and B", 20 μm. The raw data underlying panels C–E are available in S1 Data. BC, border cell; FC, follicle cell.

number of these lateral protrusions (*n* = 8) compared with controls (*n* = 6), specifically in the *mirr* region (Fig 3C, 3D, and 3F). Previous studies have shown that reducing E-cadherin (E-cad) or myosin function reduces directional persistence as a consequence of generating ectopic long side protrusions, resembling the large leading protrusion that initiate and guide BC migration [44,45]. In contrast, here we show that BCs from laminin-depleted egg chambers migrate faster towards the oocyte, despite showing an increase in the number of lateral protrusions. However, we found that the length of the ectopic side protrusions and that from the front and rear of clusters from laminin-depleted egg chambers was similar to that of controls (Fig 3K). This could explain why BCs from laminin-depleted egg chambers do not loose directionality besides having increase ectopic lateral protrusions.

As mentioned above, BCs show efficient forward movement during the early phase and increased tumbling during the late phase [39]. Next, we analyzed whether modifying the BM would affect the rotation mode. Using manual tracking (see Material and methods), we found that BC clusters from laminin-depleted egg chambers rotated slower than those from controls (*n* = 8, Fig 3G and 3G', S5 Movie). In addition, we found that while control BC clusters always rotated (*n* = 8, Fig 3K), BC clusters in laminin-depleted egg chambers did not rotate in 42.8% of the cases analyzed (*n* = 7, Fig 3K). In contrast, the slow-moving BC clusters of EHBP1

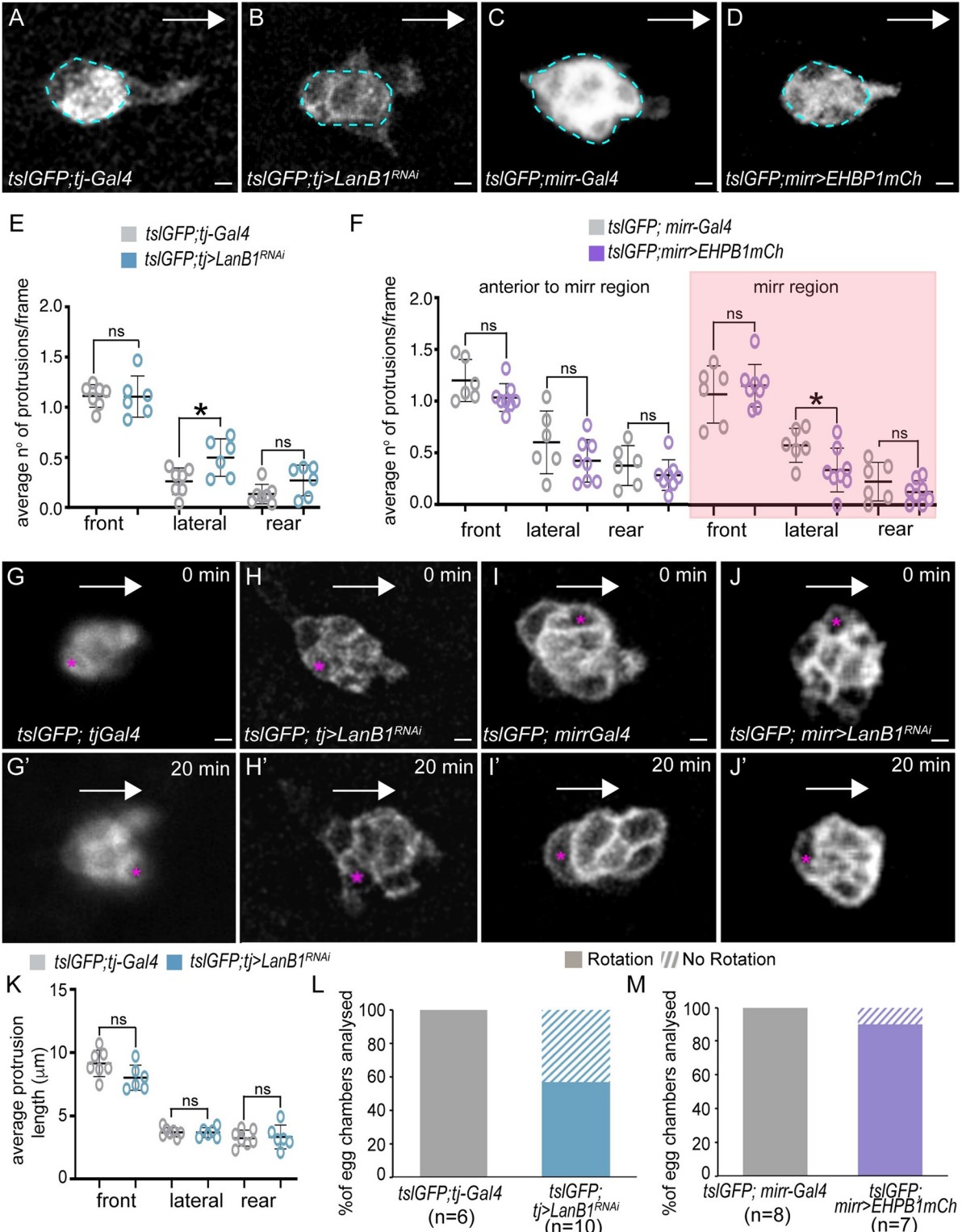

**Fig 3. Modifying BM stiffness affects protrusion orientation and migration mode.** (A–D) Stills taken from live imaging of BCs from egg chambers of the designated genotypes. The blue circles indicate the BC cluster bodies. (E, F) Mean number of protrusions observed per frame (snapshots) in each direction from BC clusters of egg chambers of the indicated genotypes. (G–J') Stills taken from live imaging of BC clusters from egg chambers of the designated genotypes. The pink asterisks mark the BC being followed. White arrows indicate the direction of the movement. (K–M) Quantification of BC cluster rotation. The statistical significance of differences was assessed with a *t* test, * *P*

value < 0.05. Horizontal and vertical lines indicate mean and SD, respectively. Scale bars in A–D and G–J, 10 μm. The raw data underlying panels E, F and K–M are available in S1 Data. BC, border cell; BM, basement membrane.

overexpressing follicles were still able to rotate (Fig 3I–3J' and 3L, $n$ = 10, S6 Movie). This result demonstrates that the stiffness of the follicle BM could alter the mode of BC migration.

## Direct correlation between BM stiffness and NC cortical tension

The current model proposes that NCs exert compressive forces over BCs that are perpendicular to the migration pathway and that mechanically influence their migration [26]. In this scenario, and since NCs are externally constrained by both the FCs and the follicle BM, the increase in migration speed we observe upon reduction of BM stiffness may arise from a drop in NC cortical tension. To test this possibility, we first analyzed to what extent stiffness of the BM could influence NC cortical tension. In wild-type egg chambers, BM stiffness over the NCs has been shown to gradually decrease from early S9 to S10 in the anterior region of the egg chamber, while a group of around 50 anterior FCs, the so-called stretched FCs (StFCs), progressively flatten in an anterior to posterior wave [46–48]. To test whether these changes in BM stiffness affected NC cortical tension, we performed laser ablation experiments in NCs under the StFCs throughout the flattening process of these cells (S5A–S5F Fig, [49]; see Materials and methods, S7 Movie). All cuts were made perpendicular to the AP axis (S5 Fig) and the behavior of cell membranes, visualized with Resille-GFP [50], was monitored up to 10 s after ablation (S7 Movie). We found that, although some differences in BM stiffness were reported for mid and late S9 egg chambers relative to early S9 [46], NC cortical tension did not change significantly throughout these stages, which is when BCs migrate ($n$ = 16, S5A–S5F Fig). However, we did detect a significant decrease in NC cortical tension at S10 ($n$ = 15, S5F Fig), when anterior FCs were totally flattened, BC migration was completed and BM stiffness was much lower than that found at earlier stages [46]. This result suggested that reducing BM stiffness below a certain threshold diminished NC cortical tension.

Based on this result, we next tested whether the decrease in BM stiffness detected in the laminin-depleted egg chambers could affect NC cortical tension. BM stiffness over the NCs was found to be homogenous along the A/P axis in early S9 control egg chambers [46,51]. In line with these results, we found, by laser ablation experiments, that NC cortical tension was also homogenous along the A/P axis at this stage (S5G Fig). Thus, we decided to perform the next set of laser ablation experiments in NCs located in the central region of early S9 follicles, ahead of migrating BCs, as this will be the environment BCs will encounter during their movement towards the oocyte (Fig 4A). We found that the cortical tension of NCs in laminin-depleted follicles (Resille-GFP-$tj$>$LanB1^{RNAi}$, $n$ = 16) was lower than in controls (Resille-GFP-$tjGal$, $n$ = 15, Fig 4B–4D, S8 Movie). Accordingly, the levels of the phosphorylated form of the regulatory chain of non-muscle Myosin II (encoded by the gene *spaghetti squash*, pSqh) at the NC cortex were lower in laminin-depleted follicles ($n$ = 10) compared with controls ($n$ = 10) (S6A, S6B and S6E Fig). An increase in NC contraction was shown to increase the levels of pSqh at the periphery of the BC cluster [26]. Here, we found that pSqh levels were reduced in the BCs of laminin-depleted egg chambers ($n$ = 9) compared with controls ($n$ = 8) (S6G, S6H and S6K Fig).

Altogether, our results support a model in which stiffness of the BM impacts on compressive forces exerted by NCs over the BCs, which is important for their movement. To further test this idea, we analyzed whether increasing BM stiffness could augment NC cortical tension. To do this, we expressed *EHBP1mCh* in the central region of the FE and measured cortical tension by laser ablation and pSqh levels. Indeed, we found that the cortical tension and pSqh

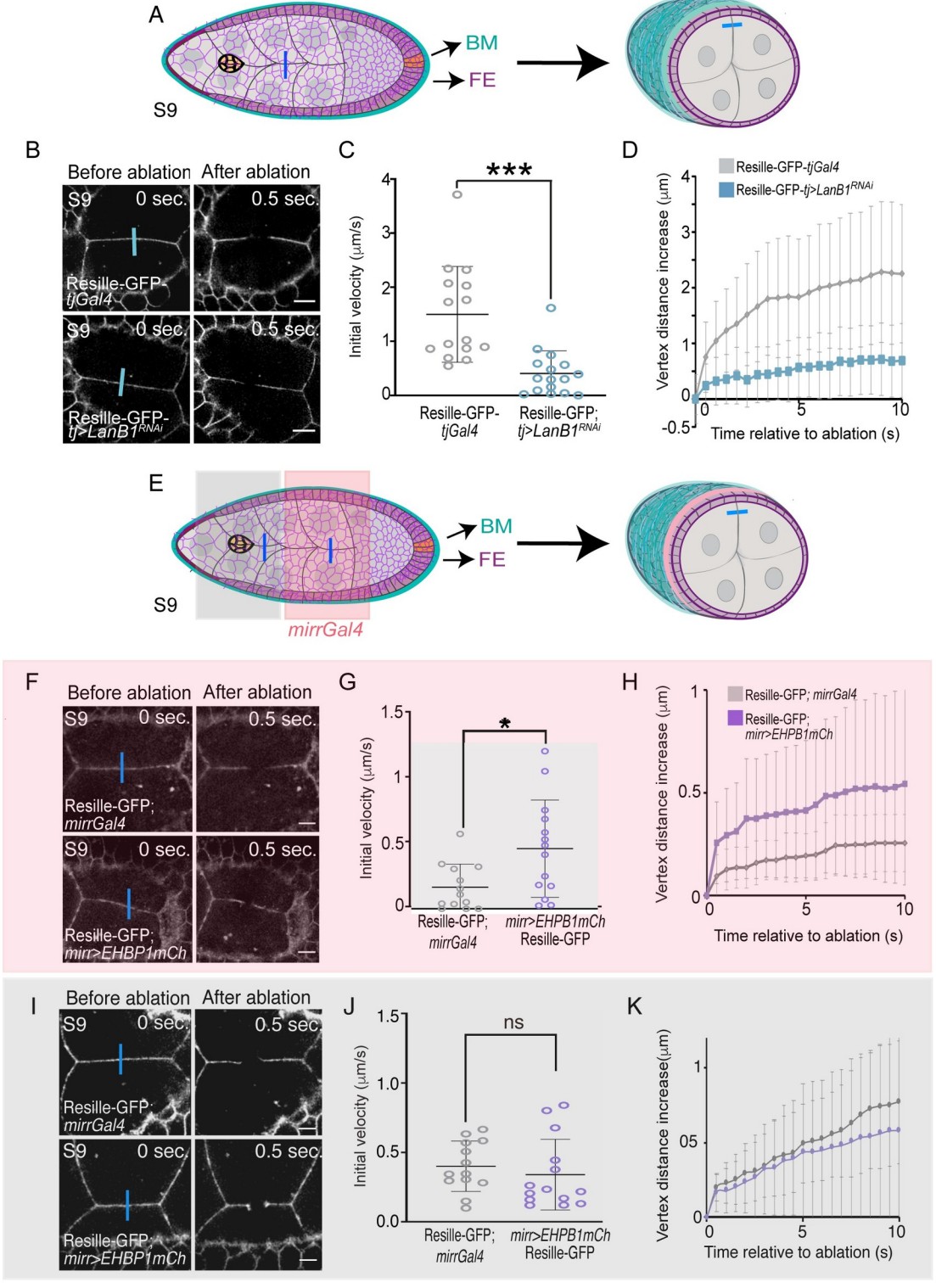

**Fig 4. Direct correlation between BM stiffness and NC cortical tension.** (A) Schematic drawings of an S9 egg chamber illustrating the point of ablation (blue bar). NCs are in gray, BCs in yellow, FCs in purple, and BM in blue. (B) Images of life S9 control (*tjGal4*) and *tj>LanB1*$^{RNAi}$ egg chambers expressing Resille-GFP before and after NC bonds are ablated. Blue bars indicate points of ablation. (C) Quantification of initial velocity of vertex displacement and (D) vertex displacement over time of the indicated ablated bonds. (E) Schematic drawing of an S9 egg chamber, similar to the one shown in A, illustrating the *mirror*Gal4 pattern of expression (*mirrGal4*, pink). (F, I) Life images of NCs located in the area anterior to (F) and at the (I) the

*mirr* expressing region of S9 control (*mirr*Gal4) and *mirr>EHBP1mCh* egg chambers expressing Resille-GFP before and after NC bonds are ablated. Blue bars indicate points of ablation. (G, J) Quantification over time of initial velocity of vertex displacement and (H, K) vertex displacement over time of the indicated ablated bonds. The statistical significance of differences was assessed with a *t* test, * *P* value < 0.05 and *** *P* value < 0.001. Horizontal and vertical lines indicate mean and SD, respectively. Scale bars in B, F, and I, 10 μm. The raw data underlying panels C, D, G, H, J, and K are available in S1 Data. BC, border cell; BM, basement membrane; FC, follicle cell; GFP, green fluorescent protein; NC, nurse cell.

levels at the cortex of NCs located under the *mirror* region in *mirr>EHBP1mCh* egg chambers (*n* = 14 and *n* = 7) was higher than that found in control (*n* = 13 and *n* = 7, Figs 4F–4H, S6C, S6D and S6F), while no statistically significant difference was observed between experimental (*n* = 14 and *n* = 7) and control (*n* = 13 and *n* = 7) NCs located anterior to the *mirror* region (Figs 4I–4K and S6F). Furthermore, we found that the levels of pSqh were higher in BCs from *mirr>EHBP1mCh* (*n* = 8, S6I, S6J, and S6L Fig), being specifically significant at the lateral cluster periphery, where contraction forces from NCs could be higher in this experimental situation.

Finally, to further test our hypothesis that BM stiffness could influence BC migration via regulating NC cortical tension, we analyzed whether a direct reduction of NC cortical tension would speed up BC migration. In order to do this, we expressed an RNAi against the Abelson interacting protein (Abi) in NCs, as its expression in FCs has been shown to cause a reduction in their cortical tension [52]. We found that the expression of an $Abi^{RNAi}$ RNAi in NCs (*n* = 5, nos>$Abi^{RNAi}$) increased BC migration speed compared with controls (*n* = 5) (S7 Fig and S9 Movie). In addition, we found that rotation of the BC cluster was affected. Thus, while BCs rotated in 100% of control *nosGal4* egg chambers (*n* = 5), rotation was hampered in 20% of *nos>AbiRNAi* egg chambers analyzed (*n* = 5, S7D Fig). This result supports our idea that less tension in NCs can prevent BC migratory rotation, which we propose contributes to increase speed.

## Contribution of cytoplasmic pressure to BC migration

Cell mechanical properties are not only dictated by cortical tension but also by cytoplasmic pressure. Collagenase treatment decreases NC cytoplasmic pressure, leading to suggest that BM stiffness could influence NC cytoplasmic pressure [51]. In this scenario, the effects on BC migration we observed when reducing laminin levels could also be due to alterations in NC cytoplasmic pressure. To test this possibility, we measured NC cytoplasmic pressure in early S9 control and *tj>LanB1^{RNAi}* egg chambers. The extent of curvature of the posterior-facing NC membrane (the radius of the circle fitting this membrane) has been proposed to reflect differences in the cytoplasmic pressure of adjacent NCs [51]. Using this method, a gradient of cytoplasmic pressure along the A/P axis, with pressure decreasing from anterior to posterior, was reported. Using this approach, we found that this gradient was maintained in both laminin-depleted and *EHBP1mCh* overexpressing follicles, as we did not detect any significant difference between the curvature of NCs from *tj>LanB1^{RNAi}* or *mirr>EHBP1mCh* follicles and that of controls (S8A–S8F Fig, *n* > 10 per position). This is consistent with previous results showing that altering the BM stiffness, by collagenase treatment, did not affect the gradient of NC cytoplasmic pressure [51]. However, even though absolute values of pressure were found lower in collagenase-treated follicles compared to untreated follicles (Lamire and colleagues) [51], we did not detect any significant difference between the curvature of NCs from *tj>LanB1^{RNAi}* or *mirr>EHBP1mCh* follicles and that of controls (S8E and S8F Fig). In any case, to more directly address a possible role of NC cytoplasmic pressure on BC migration, we analyzed BC migration in dicephalic (*dic^1*) and kelch (*Kel^{ED1}*) mutant egg chambers, as they were shown to display reduced and increased NC cytoplasmic pressure, respectively [51]. Our

in vivo analysis showed that the speed at which BCs moved in $dic^1$ and $Kel^{ED1}$ follicles was similar to that found in controls (S8G–S8J Fig and S10 Movie). All together, these results suggest that NC cytoplasmic pressure on its own does not seem to strongly contribute to the forces regulating BC migration.

## Direct correlation between BM stiffness and apical and basal FC cortical tension

As variations in BM mechanics were shown to impact on FC shape [46], we wished to determine if follicle BM stiffness affected FC cortical tension. We first checked whether reducing BM stiffness altered basal FC tension by measuring cortical tension on the basal side of FCs in S9 control and laminin-depleted FCs. Laser cuts were made in main body cuboidal FCs contacting NCs just ahead of migrating BCs and perpendicular to the dorso/ventral axis (Fig 5A and 5A'). We observed that the tension at cell–cell contacts on the basal side of FCs in laminin-depleted egg chambers (*tj*-Resille-GFP>*LanB1*$^{RNAi}$, n = 15, S11 Movie) was reduced compared to controls (*tj*-ResilleGFP, n = 15, Fig 5B–5D, S11 Movie). These results demonstrated the existence of a direct correlation between follicle BM stiffness and FC basal cortical tension. To further test this hypothesis, we assessed whether increasing BM stiffness could cause the opposite effect, a rise in basal FC tension. Indeed, we found that FCs expressing *EHBP1mCh* (Resille-GFP; *mirr*>*EHBP1mCh*, n = 17) showed increased tension at basal FC-FC contacts compared to controls (Resille-GFP; *mirr*, n = 16, Fig 5E–5G).

We next tested whether manipulating BM stiffness could also affect tension at cell–cell contacts on the apical side of FCs (S12 Movie). Recent studies have shown that decreasing Col IV levels reduced apical FC surface and caused defects in adherent junctions remodeling [46], suggesting a link between BM stiffness and FC apical tension. Similar to what happened on the basal side, our results showed that decreasing ((*tj*-Resille-GFP>*LanB1*$^{RNAi}$, n = 15) or increasing (*mirr*>*EHBP1mCh*, n = 15) BM stiffness respectively reduced or augmented tension at apical FC-FC contacts (Fig 5H–5M and S12 Movie).

All together, these results suggested that the mechanical properties of the BM could also influence FC cortical tension. Because FCs are in direct contact with NCs, FC tissue mechanics could affect BC migration. To further test this hypothesis, we analyzed whether a direct manipulation of FC tension would affect BC migration. Expression of an *Abi* RNAi in all FCs led to elimination of actin-containing networks [23,53]. Furthermore, we recently showed that a reduction in *Abi* levels rescued the increase in FC cortical tension due to integrin elimination, suggesting that *Abi* might regulate cortical tension [52]. We thus analyzed BC migration in follicles expressing an *Abi* RNAi in all FCs (*tslGFP; tj>AbiRNAi*). We found that BCs moved faster in *tslGFP; tj>AbiRNAi* follicles (n = 6) compared to controls (*tslGFP; tj*, S9 Fig and S13 Movie, n = 8). While these results supported the idea that FC tension could influence BC migration, the fact that *tjGal4* drives gene expression in all FCs, including BCs, and that *Abi* depletion can block other collective migratory processes such as egg chamber rotation [23], the defects observed in *tj>AbiRNAi* follicles could be, in part or totally, due to a specific role of *Abi* in BCs, rather than an effect on FC tension. Thus, we used *mirrGal4* to deplete *Abi* only in central FCs (*mirr>Abi*$^{RNAi}$). We first tested whether this approach altered FC tension of cell–cell contacts on the basal side of FCs. To this end, we performed laser ablation on the basal side of FCs located in the *mirror* area (Fig 6A). Indeed, we found that tension at FC-FC basal contacts was reduced in the Resille-GFP; *mirr*>*Abi*$^{RNAi}$ follicles (n = 10) compared to controls (Resille-GFP; *mirr*, Fig 6B and 6C, n = 13). Furthermore, laser ablation in NCs (Fig 6D) revealed that NC cortical tension was also reduced in the experimental egg chambers (Resille-GFP; *mirr*>*Abi*$^{RNAi}$, n = 15) compared to controls (Fig 6D–6F, n = 14). Finally, in vivo analysis

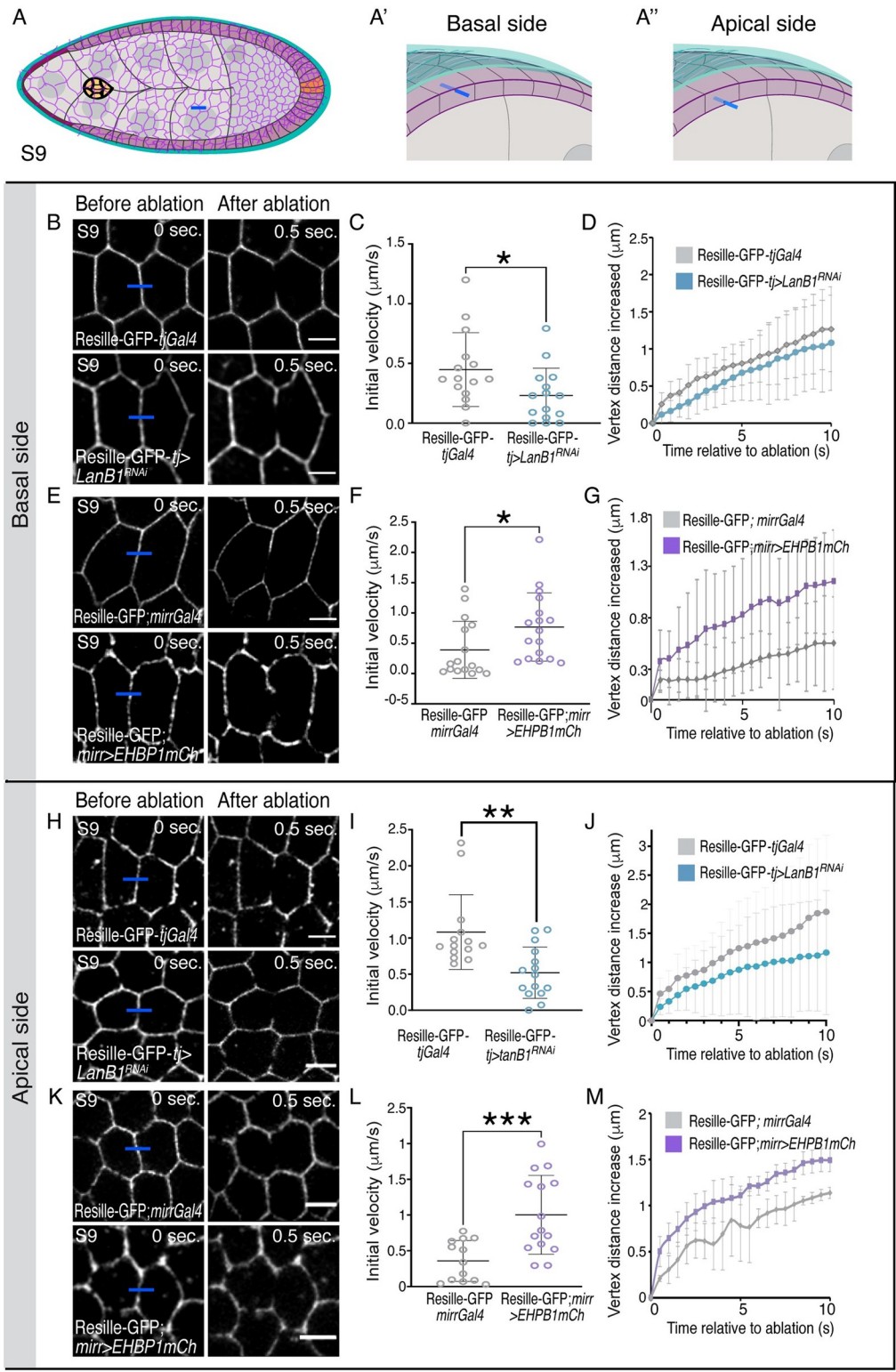

**Fig 5. Direct correlation between BM stiffness and FC cortical tension.** (A–A") Schematic drawings of an S9 egg chamber illustrating the point of ablation (blue bar) on the basal (A') and apical (A") side of a FC. NCs are in gray, BCs in yellow, FCs in purple, and BM in blue. (B, E, H, K) Images of the basal (B, E) and apical (H, K) sides of life S9 egg chambers of the indicated genotypes before and after FC bonds are ablated. Blue bars indicate points of ablation. (C, F, I, L) Quantification over time of initial velocity of vertex displacement and (D, G, J, M) vertex displacement of the

indicated ablated bonds. The statistical significance of differences was assessed with a *t* test, * *P* value < 0.05, ** *P* value < 0.01, and *** *P* value < 0.001. Horizontal and vertical lines indicate mean and SD, respectively. Scale bars in B, E, H, and K, 5 μm. The raw data underlying panels C, D, F, G, I, J, L, and M are available in S1 Data. BC, border cell; BM, basement membrane; FC, follicle cell; NC, nurse cell.

of BC migration showed that BCs migrated faster in the *mirror* region (*n* = 6) of experimental follicles compared to controls (*n* = 6) (Fig 6G–6I and S14 Movie).

## Discussion

BMs are thin, dense sheet-like types of ECM that underlie or surround virtually all animal tissues. By a combination of molecular and mechanical activities, BMs play multiple roles during tissue morphogenesis, maintenance, and remodeling [3]. One of the most prominent developmental functions attributed to BMs is cell migration. The conventional role of BMs during developmental migration is to provide an underlying substratum for moving cells. However, there are examples during morphogenesis where cells do not use BMs as a direct substratum, yet, the 3D environment through which they move is itself encased by a BM. Here, our results show that the stiffness of these encasing BMs can also influence collective cell migration, unraveling a new mechanical role for BMs during development.

The *Drosophila* egg chamber has served as a paradigm to study the role of encasing BMs in morphogenesis. Studies so far have revealed that the mechanical properties of the BM surrounding the egg chamber influence the behavior of cells in direct contact with it, the FCs. Constrictive forces applied by the egg chamber BM regulate changes in the shape of FCs that underlie tissue elongation [7,8]. BM stiffness was also shown to affect the onset and speed of the collective movement of all FCs over the inner surface of the BM, during the process of "global tissue rotation" [32]. Here, we show that the mechanical properties of the egg chamber BM can also regulate the behavior of cells that are not in direct contact with it, the BCs. BCs move using as a substratum NCs that are surrounded by FCs, encased, in turn, by a BM. Previous studies have shown that reciprocal mechanical interactions between the migrating BCs and the NCs substratum facilitate the movement of BCs [26,27]. Our results reveal that constraints exerted by the BM also influence BC migration, as stiffness of the BM evenly affects NC cortical tension. Recent studies have shown that BCs migrate along the center of the egg chamber, between the junctures where several NCs meet, as the energy cost of unzipping these junctures is most favorable [25]. In this context, we could speculate that the reduction in NC cortical tension detected when reducing BM stiffness could weaken NC-NC junctures, allowing for faster migration. In this scenario, we also propose that, in contrast to what happens in controls where the number of side protrusions is low compared to the front one, the reduced NC cortical tension we observed in laminin-depleted follicles could lead to a lessening of cellular junctions, thus explaining the increased number of side protrusions we observe in this experimental situation. However, our results also show that even though NC cortical tension is reduced in laminin-depleted egg chambers, it seems to be maintained above the threshold necessary to preserve migration along the central path. Several studies in the field have led to propose a grapple and pull model for BC migration, by forward-directed protrusions. In this scenario, BCs extend a long protrusion at the front, grip the substrate at the tip and the cell body pulls itself toward the tip, absorbing the protrusion into the cell body [54]. Furthermore, the ectopic formation of long protrusions in BCs in different experimental conditions slows BC migration [26,55]. Here, we show that BCs from laminin-depleted egg chambers show increased number of side protrusions, yet they migrate faster. However, we show that these ectopic side protrusions are not long. In fact, they have the same length as side protrusions

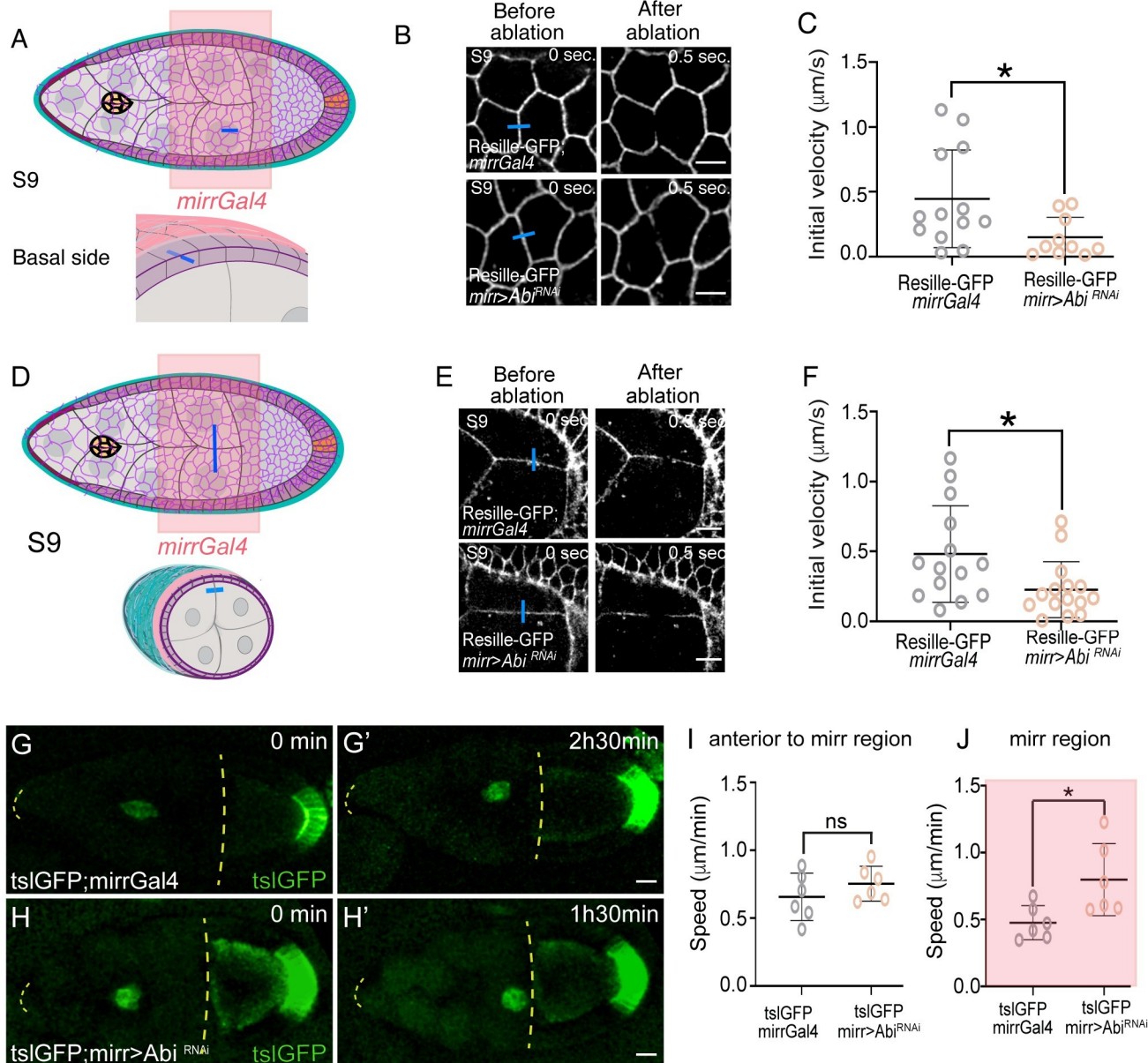

**Fig 6. Reducing the cortical tension of FCs results in reduced NCs cortical tension and increased BC migration.** (A) Schematic drawings of an S9 egg chamber illustrating the *mirror*Gal4 pattern of expression (*mirr*Gal4, pink) and the point of ablation in the basal side of FCs (blue bar). NCs are in gray, BCs in yellow, FCs in purple, and BM in blue. (B) Images of life S9 control *mirr*Gal4 and *mirr>Abi*^RNAi^ egg chambers expressing Resille-GFP before and after FCs bonds are ablated. Blue bars indicate points of ablation. (C) Quantification of initial velocity of vertex displacement of the indicated ablated bonds. (D) Schematic drawing of an S9 egg chamber illustrating the *mirrGal4* pattern of expression (pink) and the point of ablation in the NCs (blue bar). (E) Images of life S9 egg chambers of the indicated genotypes before and after NC bonds are ablated. Blue bars indicate points of ablation. (F) Quantification of initial velocity of vertex displacement of the indicated ablated bonds. (G–H') Stills taken from live imaging of migrating BCs from egg chambers of the indicated genotypes. Discontinuous yellow lines demarcate the region between the anterior border of the egg chamber and the oocyte anterior membrane. (I, J) Quantification of BC migration speed in the area anterior to (I) and at the (J) *mirr* expressing region. The statistical significance of differences was assessed with a *t* test, * P value < 0.05, ** P value < 0.01, and *** P value < 0.001. Horizontal and vertical lines indicate mean and SD, respectively. Scale bar in B, E, and G–H', 5 μm, 10 μm, and 20 μm, respectively. The raw data underlying panels C, F, I, and J are available in S1 Data. BC, border cell; BM, basement membrane; FC, follicle cell; GFP, green fluorescent protein; NC, nurse cell.

from controls. This could explain why BCs from laminin-depleted egg chambers do not loose directionality besides having increase ectopic lateral protrusions. Finally, even though the grapple and pull model is a well-accepted general model for BC motility, an alternative model

was proposed [45], in which rather than functioning as effective grapples or attachment sites, the tip of the protrusion functions as sensory organs. This was based on 3 key observations: (1) 80% of protrusions retract; (2) protruding and retracting clusters move with similar speed; and (3) BCs retract the great majority of protrusions before crawling into the space vacated by them [45]. Furthermore, BCs continue to crawl forward even though the protrusion retracts [45]. Then, they propose that "cryptic" lamellipodia observed between BCs [56] might serve a crawling function. In this context, orientation and number of protrusions do not necessarily have to correlate with migration speed, sustaining our results showing that BCs can migrate faster in laminin-depleted egg chambers, despite the increase in lateral protrusions.

Besides cortical tension, cell mechanical properties are also dictated by cytoplasmic pressure. A gradient of cytoplasmic pressure within a defined group of NCs induces TGFβ signaling in the surrounding epithelial cells. This, in turn, has been proposed to induce local softening of the BM and cell shape changes that promote elongation of the follicle, thus supporting a role for cytoplasmic pressure in shaping cells and organs [51]. However, here we show that cytoplasmic pressure on its own does not seem to compellingly influence BC migration. These results suggest that even though the different mechanical properties of cells are interconnected, each of these properties on their own might impact the diverse cell biological processes in distinct ways. In fact, cytoplasmic pressure, and not cortical tension, has been proposed to regulate *Drosophila* nurse cell shape [51]. Similarly, during the process of mitotic rounding in the columnar epithelium of the *Drosophila* wing disc, cell area expansion was shown to be largely driven by cytoplasmic pressure, while roundness was primarily driven by cell–cell adhesion and cortical stiffness [57]. In the future, it will be interesting to understand the specific contribution of the different mechanical properties to the diverse morphogenetic processes.

Altogether, we would like to propose a model of the forces regulating BC migration in which, in addition to the already described forces that NCs apply directly over BCs and vice versa, more distant compressive forces exerted by the egg chamber BM emerge as another key regulator of this migratory process, by adjusting the mechanical properties of NCs and FCs (Fig 7). Although it would be interesting to test in the future if the stiffness of the follicle BM could influence BC migration in other ways, like regulating the diffusion of chemo-attractants cues coming from the oocyte, our results prove that the 3D environment mechanically affecting developmental cell migration goes beyond adjacent cells or tissues and suggest that forces exerted by all components of this complex 3D environment are most likely coordinated to allow proper cell migration.

We believe that the new mechanical role of the BM in cell migration proposed in this work is not just restricted to BCs, as, often, during morphogenesis, migrating cells move through tissues encased by a BM. During avian cranial neural crest cell migration, the BMs from the ectoderm and neural tube separate, yet remain in close proximity forming a laminin-lined "channel" through which the cranial neural crest cells migrate [15]. Defective formation of this channel affects cranial neural crest migration [15]. We foresee that changes in the stiffness of the laminin-containing matrix lining the channel may also affect cranial neural crest migration. In the mouse cerebral cortex, BMs are found in the pia and around blood vessels [58]. Radial glia cells located near the ventricle extend long processes that attach to the pial BM. These cellular extensions serve as scaffold for the migration of Cajal–Retzius cells and neuroblasts, from the ventricular layer to the pial surface [59]. A major function of the pial BM is to provide an attachment site for the radial glia cell endfeet [59]. Damage to the pial BM results in defects in radial glial cell extension and consequently neuronal migration [59]. In this context, we believe that changes in the mechanical properties of the pial BM could also affect radial glial cell extension, which would result in neuronal migration defects.

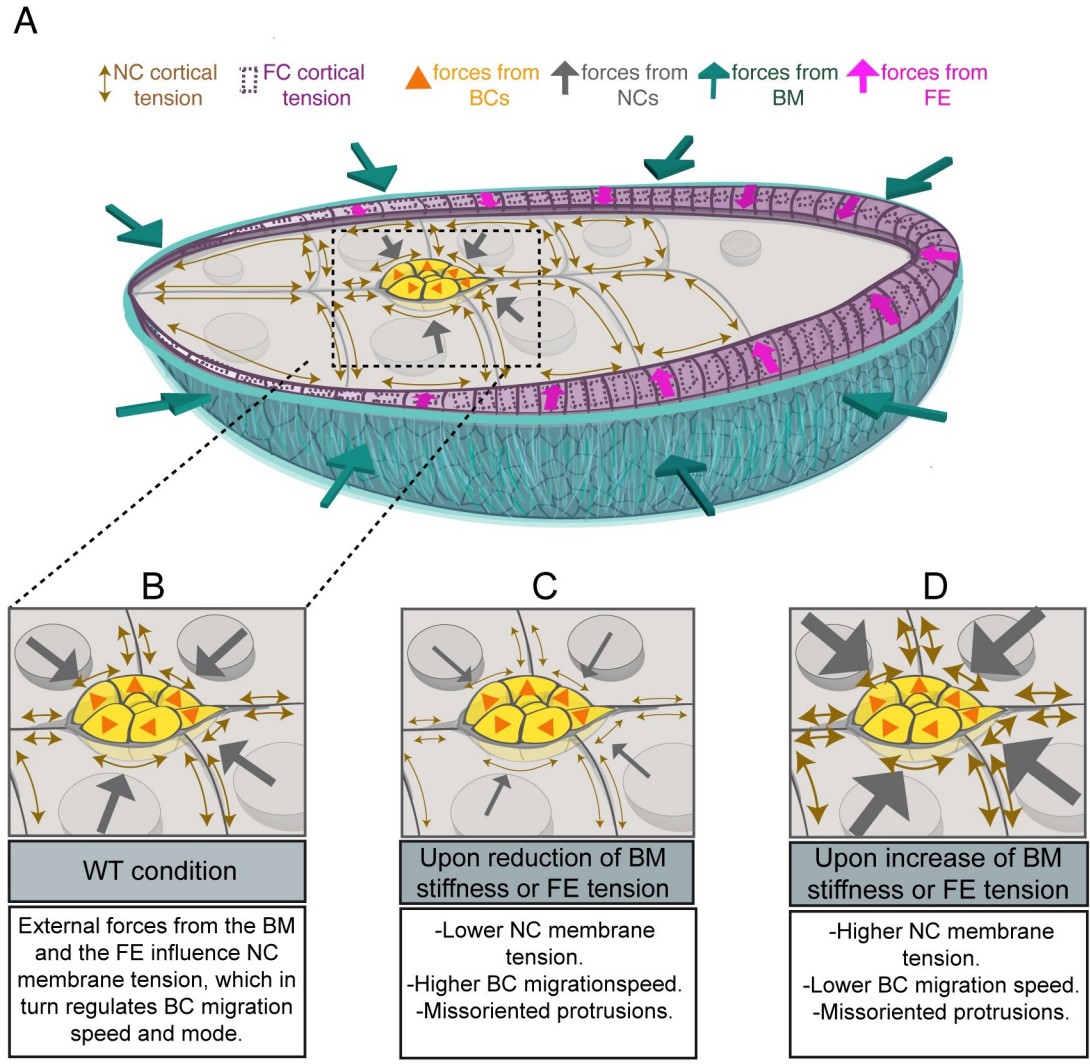

**Fig 7. Proposed model of the constrictions exerted by the BM, FCs and NCs on BCs, and their effect on BC migration.** (A) Proposed model for the forces exerted over BCs as they migrate. (B–D) Schematic drawing showing the consequences of altering the forces exerted over BCs on their migration. BC, border cell; BM, basement membrane; FC, follicle cell; NC, nurse cell.

The mechanical cues from the BMs have also been shown to contribute to the progression of several diseases including cancer. Epithelial cancer cells are bound within a BM and surrounded by a healthy stroma composed of collagen fibers and stromal cells. During the evolution of epithelial cancer invasion, cancer cells signal to the stromal cells to remodel the ECM at the tumor-stroma interface, loosen their cell–cell adhesion, and breach the BM around the tumor and invade (reviewed in [60]). Experiments over the last years have focused on the impact of the stiffness of the ECM present at the tumor-stromal interface in tumor invasion. Thus, cell culture experiments have demonstrated that the rigidity of this ECM strongly affects the migration of glioma cells [61]. Based on the results here presented, we would like to propose that the stiffness of the BM enclosing the tumor could also play an important role on the initiation of tumor invasion.

Physical constraints imposed by BMs dictate many cellular events during tissue development. This is the case of branching morphogenesis in many mammalian organs where direct cell–matrix interactions, mainly mediated by integrins, orchestrate cellular rearrangements

that sculpt the emerging tissue architecture [62]. However, not all cells undergoing developmental cellular rearrangements are in direct contact with a BM. Instead, there are many examples in which a proportion of these cells contact other cells, which themselves are enclosed by a BM. This happens during the process of budding of the mammary gland, where the BM covers multilayered terminal ends growing dynamically by cellular rearrangements [63]. Similarly, in the stratified tip of pancreas and salivary glands, both outer, in contact with the BM, and inner tip cells rearrange and change position during the process of clefting [64,65]. Based on the results we present in this work, we foresee that the mechanical properties of the BM would also influence the rearrangement of cells located in the inner layers. Investigating this will help understanding the complexity of morphogenesis in the context of multicellular developing tissues.

The results of this work unravel that, besides acting as a substratum, BMs can also regulate developmental cell migration by imposing constriction forces over the 3D environment through which cells move. A recent study, mining the Cancer Genome Atlas, has found that copy number alterations and mutations are more frequent in matrisome genes than in the rest of the genome, and that these mutations are statistically likely to have a functional impact [66]. Thus, identifying new roles for BMs in cell migration is helpful to understand not only morphogenesis but also cancer progression.

## STAR Methods

### Resource availability

**Lead contact.** Detailed information and requests for resources generated in this study should be addressed to and will be fulfilled by the lead contact María D Martín Bermudo (mdmarber@upo.es).

**Materials availability.** All reagents generated in this study are available upon request to the lead contact.

### Experimental model and subject details

***Drosophila* husbandry.** *Drosophila melanogaster* strains were raised at 25°C on standard medium. Laminin hypomorphs were trans-heterozygous for lanB1[28a] [11] and l(2)k05404 [67], neither of which affect the coding region. To knock down laminin, Coll IV or Abi levels or to increase EHBP1 expression in the FE, the *traffic jam* (*tj*) [68] and the *mirror* (*mirr*) [42] Gal4 drivers were used in combination with the following lines: *UAS-LanB1 RNAi* (VDRC 23119), *UAS-Vkg RNAi* (BDSC 50895), *UAS-Abi RNAi* (DGRC-Kyoto 9749R), and *UAS-EHBP1mch* (BDSC 67145). To visualize cell membranes in vivo, using the membrane marker Resille-GFP, and, at the same time, use the *tj* Gal4 driver to express a desired UAS construct in all FCs, we generated flies with a recombinant second chromosome carrying both transgenes, Resille-GFP-*tj*. Stocks used in this study are described in Table 1.

**Generation of *Drosophila* transgenic lines expressing GFP in border cells.** To generate the *tslGFP* construct, an EcoR1/BamH1 fragment from C4PLZ-*tsl*(G), previously shown to drive expression in BCs [38], was subcloned into the pCasper-GFP-PH vector [69]. The plasmid was introduced into the germ line of *w*[1118] flies by the company BestGene following standard methods and several independent transgenic lines were isolated.

### Methods details

**Immunohistochemistry.** Flies were grown at 25°C and yeasted for 2 days prior to dissection. Ovaries were dissected at room temperature in Schneider's medium (Sigma Aldrich).

After that, fixation was performed incubating egg chambers for 20 min in 4% paraformaldehyde in PBS (ChemCruz). Samples were then permeabilized with a phosphate-buffered saline +1% Tween20 solution (PBT) and incubated in a blocking solution (PBT10) for 1 h. Ovarioles were then incubated with the primary antibodies, rabbit anti-phosphomyosin (Cell Signalling) or anti-LanB1 antibody (Abcam, EPR3189) overnight, followed by a washing step and incubation with secondary antibodyies Cy3 and Cy2 (Jackson ImmunoReseach Laboratories). To visualize F-acting and DNA, ovarioles were incubated for 30 min with Rhodamine Phalloidin (Molecular Probes, 1:40) and 10 min with Hoechst (Molecular Probes, 1:100), respectively. Experimental and control ovarioles were always mixed and stained together. Samples were mounted in Vectashield (Vector Laboratories) and imaged on a Leica SPE confocal microscope (DM 2500).

**Time-lapse imaging.** For time-lapse imaging, 1 to 2 days old flies were fattened on yeast for 24 to 48 h before dissection. Ovaries were dissected in Schneider's medium. Culture conditions were performed as described in [43]. Movies were acquired on a Leica SP5 MP-AOBS confocal microscope with a 40 × 1, 3 PL APO objective and a Nikon confocal microscope with a 40× oil objective. Z stacks of 30 to 40 slices (0.75 to 1.5 mm interval) were taken every 3 to 4 min.

**Atomic force microscopy measurements.** Atomic force microscopy measurements were performed as in [32]. Ovarioles were dissected out of the muscle sheath to make sure that the AFM cantilever was in direct contact with the BM. Stiffness of ex vivo ovarian tissues immobilized on Petri dishes using Cell-Tak (BD Biosciences, Oxford, United Kingdom) and containing *Drosophila* culture media was tested by AFM within 30 min of dissection. Monodisperse polystyrene beads (diameter 5.46 ± 0.12 μm, microParticles GmbH, Berlin, Germany) were glued to a silicon cantilever with a nominal spring constant of 0.03 N/m (Arrow TL1, Nano-World, Neuchatel, Switzerland). Cantilevers were mounted on a JPK CellHesion 200 (Bruker Nano, Berlin, Germany) set up on an inverted optical microscope. Exact cantilever spring constant was determined using the thermal noise method included in the AFM software. The cantilever was positioned at the desired position by brightfield microscopy. Force-distance-curves were taken at an approach speed of 10 μm/s and a maximum force F = 5 nN. Force–distance curves were analyzed using the JPK Data Processing software, which fits the Hertz model to the data. The apparent elastic modulus, $K = E/1\text{-}v^2$, which is a measure of elastic stiffness, was determined using $F = 4/3 \, K \sqrt{R}\delta^{3/2}$, where R is the radius of the indenter, δ the indentation depth, and F the applied force, with E being the Young's modulus and ν the Poisson's ratio. Normal distribution of AFM measurements was confirmed using the Kolmogorov–Smirnov test. The statistical significance of the differences between experimental and control values was evaluated using two-tailed *t* tests, with *p* value thresh-holds of $^{*}p < 0.05$, $^{**}p < 0.01$, and $^{***}p < 0.001$.14 measurements from 6 control egg chambers, 16 from 8 laminin-depleted, 15 from 7 Col IV-depleted, and 14 from 7 EHPB1mCh overexpressing ovarioles were collected. In addition, between 4 to 6 measurements in both the anterior and *mirror* region from 4 *mirr* and *mirr> EHPB1mCh* different ovarioles were collected.

**Laser ablation.** Laser ablation experiments were performed in an Olympus IX-81 inverted microscope equipped with a spinning disk confocal unit (Yokogawa CSU-X1), a 100× oil objective, a 355 nm pulsed laser, third-harmonic, solid-state UV laser, and an Evolve 512 EMCCD digital camera (Photometrics). To analyze tension in NCs and FCs, a pulse of 1,000 mJ energy and 20 msec. and 75 mJ energy and 4 msec. duration, respectively, was applied to sever plasma membranes of cells. In all cases, cell surfaces were visualized with the membrane marker Resille-GFP and a Cobolt Calypso state laser (l = 491 nm 50 mW) was used for excitation of the GFP. To minimize potential effects due to anisotropic distribution of forces in NCs and FCs, cuts were made perpendicular to the AP axis and to the dorsal ventral axis of the egg

chamber, respectively. Images were taken 3 s before and 10 s after laser pulse, every 0,5 s. To analyze the vertex displacements of ablated cell bonds, the vertex distance increase from different ablation experiments (DL) was averaged using as $L_0$ the average of distance of the vertexes 3 s before ablation. The initial velocity was estimated as the velocity at the first time point (t1 = 0,5 sec). Standard deviation (SD) was determined.

**Image processing and data analysis.** Measurements of cluster velocity, cluster rotation, and protrusion dynamics were done manually as described in [70]. Briefly, once the body of the cluster was defined according to [40], forward directed speed was calculated using the distance between the center of the cluster at 1 time point and the next in the X axis only. When needed, drift of egg chambers was corrected using the plugin StackReg from FIJI-Image J [71]. To study BC cluster rotation, the position of an individual BC was manually tracked in z stacks of GFP confocal sections time point by time point for an interval of 20 min during the late phase of migration. To analyze protrusion orientation, a line was drawn from the center of the cluster to the tip of the extension. The angle of this line relative to the x-axis was calculated. Extensions were classified as front (45˚ to 315˚), side (255˚ to 315˚ and 45˚ to 135˚), and back (135˚ to 225˚). In all cases, an average of 150 protrusions was analyzed.

Analysis of the radius of curvature of NCs was carried out by measuring the radius of manually drawn circles that fit the curvature of the posterior facing membranes of NCs. This quantification was done only on 1 NC per row (anterior, central anterior, central posterior, and posterior) in at least 10 independent follicles.

For the quantification of pSqh fluorescent intensities in NCs, square regions of interest (ROIs) of 28.94 $\mu m^2$ (20 × 20 pixels) were applied to the membranes of central-posterior and posterior NCs. The mean gray value was then measured using the FIJI-Image J measure tool. This analysis was performed in individual Z-confocal sections corresponding to focal planes in which BCs were visible. For the quantification of pSqh fluorescence intensity in BCs, maximum projections of 5 Z-confocal sections were taken every 0.75 μm. Then, 28.94 $\mu m^2$ square ROIs were applied to the front, rear, and both sides of BCs and the mean gray value was then measured with the FIJI-Image J measure tool.

Statistical analysis of significant differences between control and experimental samples was done using Student's *t* test, * *P* value < 0.05, ** *P* value < 0.01, and *** *P* value < 0.001. Outliers from data sets were identified by the ROUT method with Q = 1% and were not included in the analysis and graphs.

## Supporting information

**S1 Movie. In vivo BC migration in control and laminin- and Col IV-depleted egg chambers.** BC migration in control (*tsl*GFP; *tj*Gal4) and laminin (*tsl*GFP; *tj*>*LanB1^RNAi*, *tsl*GFP; *Lanb1^hyp*) or Col IV (*tsl*GFP;*tj*>*Col IV RNAi*)-depleted egg chambers, related to Fig 1. Scale bar, 20 μm.
(AVI)

**S2 Movie. In vivo migration of control and *LanB1^RNAi* expressing BCs.** Migration of control (*C306; slbo; tslGFPGal4*) and *LanB1^RNAi* expressing BCs (*C306;slbo;tsl;GFP>LanB1 RNAi*), related to S3 Fig. Scale bar, 20 μm.
(AVI)

**S3 Movie. In vivo BC migration in control and *tj*>*EHBP1mCh* egg chambers.** BC migration in *tsl*GFP; *tj*Gal4 and *tsl*GFP; *tj*>*EHBP1mCh* egg chambers, related to S3 Fig. *tsl*GFP is in green and *EHBP1mCh* in red, related to S4 Fig. Scale bar, 20 μm.
(AVI)

**S4 Movie. In vivo BC migration in control and *mirr>EHBP1mCh* egg chambers.** BC migration in *tsl*GFP; *mirrGal4* and *tsl*GFP; *mirr>EHBP1mCh* egg chambers. *tsl*GFP is in green and *EHBP1mCh* in red, related to Fig 2. Scale bar, 20 μm.
(AVI)

**S5 Movie. BC cluster rotation in control and laminin-depleted egg chambers.** BC cluster rotation in control (*tsl*GFP; *tjGal4*) and laminin-depleted (*tsl*GFP; *tj>LanB1RNAi*) egg chambers, related to Fig 3. Scale bar, 10 μm.
(AVI)

**S6 Movie. BC cluster rotation in control and EHBP1mCh overexpressing egg chambers.** BC cluster rotation in control (*tsl*GFP; *mirrGal4*) and *EHBP1mCh* overexpressing (*tsl*GFP; *mirr>EHBP1mCh*) egg chambers, related to Fig 3. Scale bar, 10 μm.
(AVI)

**S7 Movie. Laser ablation of cell bonds between NCs of control egg chambers.** Movies correspond to the ablation experiment shown in S5 Fig NCs membranes are visualized with Resille-GFP. A cell bond between 2 control NCs is ablated. GFP fluorescent is lost in the middle of the ablated bond upon laser ablation. The movie continues 10 s after the cut and shows displacement of the vertexes. Images are taken every 0.5 s. Scale bar, 10 μm.
(AVI)

**S8 Movie. Laser ablation of cell bonds between NCs of control and laminin-depleted egg chambers.** Movies correspond to the ablation experiment shown in Fig 4. NCs membranes are visualized with Resille-GFP. A cell bond between 2 control NCs is ablated. Upon laser ablation, GFP fluorescent is lost in the middle of the ablated bond. The movie continues 10 s after the cut and shows displacement of the vertexes. Images are taken every 0.5 s. Scale bar, 10 μm.
(AVI)

**S9 Movie. In vivo BC migration in control and *nos>abiRNAi* chambers.** BC migration in *nos;his*YFP and *nos>AbiRNAi;his*YFP egg chambers, related to S7 Fig. Scale bar, 20 μm.
(AVI)

**S10 Movie. In vivo BC migration in control and in *dic¹* and *Kel^{ED1}* mutant egg chambers.** BCs were directly visualized using bright field, related to S8 Fig. Scale bar, 20 μm.
(AVI)

**S11 Movie. Laser ablation on the basal side of cell bonds between FCs of control and laminin-depleted egg chambers.** Movies correspond to the ablation experiment shown in Fig 5. The membranes on the basal side of FCs are visualized with Resille-GFP. A cell bond between 2 control FCs is ablated. GFP fluorescent is lost in the middle of the ablated bond upon laser ablation. The movie continues 10 s after the cut and shows displacement of the vertexes. Images are taken every 0.5 s. Scale bar, 5 μm.
(AVI)

**S12 Movie. Laser ablation on the apical side of cell bonds between FCs of control and laminin-depleted egg chambers.** Movies correspond to the ablation experiment shown in Fig 5. The membranes on the apical side of FCs are visualized with Resille-GFP. A cell bond between 2 control FCs is ablated. GFP fluorescent is lost in the middle of the ablated bond upon laser ablation. The movie continues 10 s after the cut and shows displacement of the vertexes. Movie length and frame rate are as described for Movie S11. Scale bar, 5 μm.
(AVI)

**S13 Movie. In vivo BC migration in control and *tj>AbiRNAi* chambers.** BC migration in *tsl*GFP; *tjGal4* and *tsl*GFP; *tj>AbiRNAi* egg chambers. Scale bar, 20 μm.
(AVI)

**S14 Movie. In vivo BC migration in control and *mirr>abiRNAi* chambers.** BC migration in *tsl*GFP; *mirrGal4* and *tsl*GFP; *mirr> AbiRNAi* egg chambers, related to Fig 6. Scale bar, 20 μm.
(AVI)

**S1 Fig. Quantification of laminin levels in S9/S10 *laminin*-depleted ovaries.** (A) S10 control *tjGal4* and (B) *tj>LanB1RNAi* (B) egg chambers stained with anti-LanB1 antibody (green), the DNA marker Hoechst (blue) and the F-actin marker Rhodamine-Phalloidin (F-actin, red). (C) Quantification of the LanB1 levels in egg chambers of the specified genotypes. The statistical significance of differences was assessed with a *t* test, *** *P* value < 0.001. Horizontal and vertical lines indicate mean and SD, respectively. Scale bars in A and B, 20 μm. The raw data underlying panel C are available in S1 Data.
(JPG)

**S2 Fig. Reducing laminin levels in FCS results in a decrease in BM stiffness.** (A) Schematic drawing of an early S9 egg chamber illustrating the BCs (yellow), NCs (gray), FCs (purple), BM (green), and the position where AFM measurements were taken (arrow). (B) Comparison of the apparent elastic modulus *K* in egg chambers of the designated genotypes. (C) Schematic drawing of a middle S9 egg chamber illustrating the BCs (yellow), NCs (gray), FCs (purple), BM (green), the mirror region (pink square) and the positions where AFM measurements were taken, anterior to the *mirr* region (A, black arrow) and in the *mirr* region (C, pink arrow). (D, E) Comparison of the apparent elastic modulus *K* in the *mirr* region (C) and anterior to the *mirr* region (E), in 4 control *mirrGal4* (D) and *mirr>EHBP1mCh* egg chambers. A minimum of 4 different readings (circles)/egg chamber were taken. Horizontal lines in B, D, and E represent mean values. The statistical significance of differences was assessed with a *t* test, * *P* value < 0.05, ** *P* value < 0.01, and *** *P* value < 0.001. Horizontal and vertical lines indicate mean and SD, respectively. The raw data underlying panels B, D, and E are available in S1 Data.
(JPG)

**S3 Fig. Decreasing laminin levels in BCs does not affect their migration.** (A, A') S9 *triple>LanB1RNAi* (B) egg chambers stained with anti-LanB1 antibody (green), the DNA marker Hoechst (blue) and the PC marker Fasciclin III (FasIII, red). (B–C') Stills taken from live imaging of BCs from control (*C306Gal4; slboGal4; tslGal4*) and *C306; slbo; tsl>LanB1^{RNAi}* egg chambers. (D) Quantification of the migration defects in egg chambers of the indicated genotypes. The statistical significance of differences was assessed with a *t* test. Horizontal and vertical lines indicate mean and SD, respectively. Scale bars in A" and B", 20 μm. The raw data underlying panel D are available in S1 Data.
(JPG)

**S4 Fig. Elevating EHBP1 levels in all FCs results in reduced BC migration speed.** (A–B') Stills taken from live imaging of migrating BCs from egg chambers of the indicated genotypes. Scale bar in A' and B', 20 μm.
(JPG)

**S5 Fig. NC cortical tension throughout egg chamber development from early S9 to S10.** (A–D) Schematic drawings of early S9 (A), middle S9 (B), late S9 (C), and S10 egg chamber illustrating the BCs (yellow), NCs (gray), FCs (purple), BM (green), and the point of ablation in the NCs (blue bar). (A'–D') Images of life control egg chambers of the indicated

developmental stages expressing Resille-GFP before and after NC bonds are ablated. Blue bars indicate points of ablation. (E) Schematic representation of an egg chamber and the point of ablation in NCs. (F, G) Quantification of the initial velocity of vertex displacement of the indicated ablated bonds. The statistical significance of differences was assessed with a *t* test, * *P* value < 0.05 and ** *P* value < 0.01. Horizontal and vertical lines indicate mean and SD, respectively. Scale bar in A'–D', 10 μm. The raw data underlying panels F and G are available in S1 Data.
(JPG)

**S6 Fig. Modifying BM stiffness affects pSqh levels at the cortex of NCs and BCs.** (A–D, G–J') S9 egg chambers of the designated genotypes stained with anti-pSqh (red) and the DNA marker Hoechst (blue) showing pSqh levels in the NCs (A–D) and the BCs (G–J'). (E, F, K, L) Quantification of pSqh levels in the NCs (E, F) and in the BCs (K, L) of egg chambers of the specified genotypes. G'–J' Magnifications of the white boxes in G–J. The statistical significance of differences was assessed with a *t* test, * *P* value < 0.05, ** *P* value < 0.01, and *** *P* value < 0.001. Horizontal and vertical lines indicate mean and SD, respectively. Scale bar in A, 20 μm. The raw data underlying panels E, F, K, and L are available in S1 Data.
(JPG)

**S7 Fig. Decreasing *abi* RNA levels in NCs results in increased BC migration speed.** (A–B') Stills taken from live imaging of migrating BCs from egg chambers of the indicated genotypes. (C) Quantification of the migration defects in egg chambers of the specified genotypes. The statistical significance of differences was assessed with a *t* test, *** *P* value < 0.001. Horizontal and vertical lines indicate mean and SD, respectively. Scale bars in A' and B', 20 μm. The raw data underlying panel C and D are available in S1 Data.
(JPG)

**S8 Fig. Contribution of NC cytoplasmic pressure to BC migration.** (A–D) S9 controls *tjGal4, tj>LanB1^{RNAi}, mirrGal4* and *mirr>EHBP1mCh* egg chambers stained with the F-actin marker Rhodamine-Phalloidin (F-actin, red) and the DNA marker Hoechst (blue). Arrows in A indicate the curvature of the membranes between an anterior (A) and an anterior central NC (Ca); between 2 central NCs -a Ca and a Cp (a posterior central NC); between a Cp and a posterior (P) NC and between a P NC and the oocyte membrane. (E, F) Quantification of the radius of curvature of A, Ca, Cp and P NCs in S9 egg chambers of the indicated genotypes. BC clusters are marked with a red circle. (G–I) Stills taken from live imaging of migrating BCs from egg chambers of the indicated genotypes. (J) Quantification of BC migration speed in egg chambers of the indicated genotypes. The statistical significance of differences was assessed with a *t* test. Horizontal and vertical lines indicate mean and SD, respectively. Scale bars in A–D and G–I', 20 μm. The raw data underlying panels E, F, and J are available in S1 Data.
(JPG)

**S9 Fig. Decreasing *abi* RNA levels in all FCs results in increased BC migration speed.** (A–B') Stills taken from live imaging of migrating BCs from egg chambers of the indicated genotypes. (C) Quantification of the migration defects in egg chambers of the specified genotypes. The statistical significance of differences was assessed with a *t* test, *** *P* value < 0.001. Horizontal and vertical lines indicate mean and SD, respectively. Scale bars in A' and B', 20 μm. The raw data underlying panel C are available in S1 Data.
(JPG)

**S1 Data. Raw data behind all graphs.**
(XLSX)

## Acknowledgments

We acknowledge the Bloomington Stock Centre and the Developmental Studies Hybridoma Bank for fly stocks and antibodies. We also thank Sara Martín Villanueva and Marc Furriols for their help in generating the *tslGFP* construct and transgenic flies. Finally, we are grateful to A. González-Reyes for helpful remarks on the manuscript.

## Author Contributions

**Conceptualization:** Ester Molina López, Anna Kabanova, María D. Martín-Bermudo.

**Formal analysis:** Ester Molina López, Anna Kabanova, María D. Martín-Bermudo.

**Funding acquisition:** María D. Martín-Bermudo.

**Investigation:** Ester Molina López, María D. Martín-Bermudo.

**Methodology:** Ester Molina López, Anna Kabanova, Alexander Winkel, Kristian Franze, Isabel M. Palacios, María D. Martín-Bermudo.

**Project administration:** María D. Martín-Bermudo.

**Resources:** María D. Martín-Bermudo.

**Supervision:** María D. Martín-Bermudo.

**Validation:** Ester Molina López, Anna Kabanova, María D. Martín-Bermudo.

**Writing – original draft:** María D. Martín-Bermudo.

**Writing – review & editing:** Ester Molina López, Anna Kabanova, Isabel M. Palacios, María D. Martín-Bermudo.

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
