## [Editor Report · Decision Letter 0]

23 Mar 2023

Dear Dr Martin-Bermudo, 

Thank you for submitting your manuscript from Review Commons entitled "Constriction forces imposed by basement membranes regulate developmental cell migration" for consideration as a Research Article by PLOS Biology. Please accept my apologies for the delay in getting back to you as we consulted with an academic editor about your submission.

Your revised manuscript and rebuttal has now been evaluated by the PLOS Biology editorial staff, as well as by an academic editor with relevant expertise, and I am writing to let you know that we would like to send your submission for re-review by the original reviewers at Review Commons.

However, before we can send your manuscript back to the reviewers, we need you to complete your submission by providing the metadata that is required for full assessment. To this end, please login to Editorial Manager where you will find the paper in the 'Submissions Needing Revisions' folder on your homepage. Please click 'Revise Submission' from the Action Links and complete all additional questions in the submission questionnaire.

Once your full submission is complete, your paper will undergo a series of checks in preparation for peer review. After your manuscript has passed the checks it will be sent out for review. To provide the metadata for your submission, please Login to Editorial Manager (https://www.editorialmanager.com/pbiology) within two working days, i.e. by Mar 25 2023 11:59PM.

Kind regards,

Richard

Richard Hodge, PhD

Associate Editor, PLOS Biology

rhodge@plos.org

PLOS

---

## [Decision Letter · Decision Letter 1]

27 Apr 2023

Dear Dr Martin-Bermudo,

Thank you for your patience while we considered your revised manuscript "Constraints imposed by basement membranes regulate developmental cell migration" for consideration as a Research Article at PLOS Biology. Please accept my apologies for the delay in getting back to you this week. Your revised study has now been evaluated by the PLOS Biology editors, the Academic Editor and the original reviewers at Review Commons.

In light of the reviews, which you will find at the end of this email, we are pleased to offer you the opportunity to address the remaining points from the reviewers in a revision that we anticipate should not take you very long. We will then assess your revised manuscript and your response to the reviewers' comments with our Academic Editor aiming to avoid further rounds of peer-review, although might need to consult with the reviewers, depending on the nature of the revisions.

In addition, I would be grateful if you could please address the following editorial and data-related requests that I have provide below:

(A) We would like to suggest the following modification to the title (as suggested by Reviewer #2):

““Constriction Imposed by Basement Membrane Regulates Developmental Cell Migration”

(B) You may be aware of the PLOS Data Policy, which requires that all data be made available without restriction: http://journals.plos.org/plosbiology/s/data-availability. For more information, please also see this editorial: http://dx.doi.org/10.1371/journal.pbio.1001797

-Supplementary files (e.g., excel). Please ensure that all data files are uploaded as 'Supporting Information' and are invariably referred to (in the manuscript, figure legends, and the Description field when uploading your files) using the following format verbatim: S1 Data, S2 Data, etc. Multiple panels of a single or even several figures can be included as multiple sheets in one excel file that is saved using exactly the following convention: S1_Data.xlsx (using an underscore).

-Deposition in a publicly available repository. Please also provide the accession code or a reviewer link so that we may view your data before publication. 

Figure 1H-J, 2C-D, 3E-F, 3K-L, 4C-D, 4G-H, 4J-K, 5C-D, 5F-G, 5I-J, 5L-M, 6C, 6F, 6I-J, S1C, S2B, S2D-E, S3D, S5F-G, S6E-F, S6K-L, S7C, S8E-F, S8J, S9C 

(C) Please also ensure that each of the relevant figure legends in your manuscript include information on *WHERE THE UNDERLYING DATA CAN BE FOUND*, and ensure your supplemental data file/s has a legend.

(D) Please ensure that your Data Statement in the submission system accurately describes where your data can be found and is in final format, as it will be published as written there. 

As you address these items, please review your reference list to ensure that it is complete and correct. If you have cited papers that have been retracted, please include the rationale for doing so in the manuscript text, or remove these references and replace them with relevant current references. Any changes to the reference list should be mentioned in the cover letter that accompanies your revised manuscript.

**IMPORTANT - SUBMITTING YOUR REVISION**

*Resubmission Checklist*

*Published Peer Review*

*PLOS Data Policy*

*Blot and Gel Data Policy*

Sincerely,

Richard

Richard Hodge, PhD

Associate Editor, PLOS Biology

rhodge@plos.org

REVIEWS:

Reviewer #1: The revised manuscript by Lopez et al is much better specially with the inclusion of Atomic force microscopy data. The authors have addressed most of my concerns satisfactorily. 

However, I do have some concern regarding the choice of word in the manuscript and seek some clarifications listed below. 

1. I have a concern regarding the word "regulate" in the title, abstract and text. Since the basement membrane is affecting the kinetics of border cell migration by modulating the cortical tension of nurse cell and follicle cells, I would believe this effect is indirect. Thus it would be appropriate to use the word "affect" or "modulate" in the title, abstract and text over the word "regulate." 

The title may read as "Constraints imposed by basement membranes modulates developmental cell migration. "

Some minor comments: 

1. I do seek some clarification in Supplementary Figure 2D and 2E. What do

C1, A1; C2, A2; C3, A3 and C4, A4 denote? Are these a set of readings for different points in region C and A? Secondly, for each point like C1 are the circles denoting one reading? Figure legend can include more information to render clarity on this.

Secondly in the materials & methods section under "Atomic Force Microscopy measurements" the authors are using the word ovariole (Line 765-768) instead of egg chamber. 

2. Line 225 "Thus, even though Laminins are not expressed in.." Delete word "Thus"

Reviewer #2 (Andrea Page-McCaw, signs review): I think the significance of this work warrants its publication in PLoS Biology, as it reveals a new, important, and generalizable mechanism by which the stiffness of basement membranes influences cell migration. 

I had very few requests on the previous round of review at Review Commons, and I am generally satisfied that the paper is ready for publication. I have a few points for the authors and editor to consider together:

1. I recognize that the authors changed the title at my request, but I think "constraint" is too vague. How about "Constriction Imposed by Basement Membrane Regulates Developmental Cell Migration"

2. Give the sample number in Fig. 3K,L on the figure or in the legend.

3. There are still some cases when I wish the authors were specific about the effect, rather than saying "affected" and "influenced". Examples are lines 213, 217, 293, 295 (line numbers from the marked up manuscript)

4. Although the developmental changes in cortical tension are interesting, they do not demonstrate that BM affects that cortical tension - there could be many changes happening in development. The direct manipulations of the BM are sufficient to address this question. I suggest softening the word demonstrated in line 317.

5. The experiments on cytoplasmic pressure should be presented without so much attention on the A-P gradient of pressure and more on the absolute pressure. The A-P gradient of pressure is not oriented correctly to explain either BC migration or to transmit the constriction of the basement membrane; however the absolute pressure does both. I don't find the radius of curvature to be relevant because it only reports the difference in pressure. The relevant experiments here are dic and kel mutants, which Lamire et al 2020 showed to have reduced pressure by about 4-fold (dic) and increased pressure by about 30% (kel). The authors do not find a change in BC speed in these mutants, and that is relevant. But I still think the authors should be just a bit cautious about their interpretation, considering that Lamire et al 2020, defined cytoplasmic pressure as "the force that counters cortical tension". Since Martín-Bermudo and colleagues show that cortical tension is modulated by the basement membrane and is important for BC migration, it appears on the face of it that cytoplasmic pressure would also be important, and Lamire did find that collagenase reduced cytoplasmic pressure of nurse cells by about 6-fold.

Reviewer #3: Ester and her colleagues tried to address our comments from their previous manuscript. However, their discussion seems not be to supportive while ignoring some previous findings published by other teams. We feel that main conclusion of this manuscript might be true, but the precise mechanism linking the phenotypes (from matrix stiffness to cell tension to border cell migration) might be not precise or fully understood. Thus, we recommend that new submission can be considered again if authors can really address our worried points, as shown below:

Major comments:

1. The main reference paper (Mishra et al., 2019) in the revised discussion demonstrated that myosin works with E-cadherin to communicate the direction from the lead cell to the followers and that the non-autonomous function and the autonomous function within protrusions need to cooperate to stop follower cells from protruding. This means the lateral protrusions need to be restricted for efficient BC migration, whether they grapple or not. So, main conclusion of Mishra's paper contradicted the hypothesis of currently revised manuscript that lower NC tension leads to a lessening of cellular junctions, which increases the frequency of lateral protrusions, thus leading to a faster BC migration speed. 

2. We also don't agree with authors about the biological role of BC protrusions as a sensory organ, since this discussed point will fight against the traditional function of E-cadherin adhesion between BCs and surrounding NCs. Several papers in the field strongly demonstrated that BC-NC E-cadherin adhesion is permissive for BC migration through between NCs, thus supporting that "grappling" structure should be present in BCs to allow their transient anchorage on the surface of surrounding NCs. Thus, where is "grappling" structure in border cells, if this is not protrusion of BCs? Compared with main body regions of BCs, protrusions have more trend to behave as a "grappling" structure for BC anchorage on NCs.

3. Authors demonstrated that less NC tension leads to more large protrusions in BCs, which is apparently opposite to some findings of two previous paper (Aranjuez et al., 2016; Lamb et al., 2021): more tension by either RhoGEF expression or Singed loss in NCs can strongly result in an increase of p-MRLC and tension in NCS, thereby leading to enhanced protrusion formation in multiple BCs and slowing BC migration. So, how do authors comment on these conclusions from both published papers to explain the major conclusion (less NC tension results in multiple protrusion formation and faster speed)?

4. About the tension property of NCs, corset property of external follicle cells should be present in apical and later basal medial region, which has also been shown in (Balaji et al., 2019) to contract apical surface of FCs to resist the expansion of internal germ cells. However, authors did laser ablation in FC junctions, but not at apical and basal medial region. Can authors explain the reason of choosing this subcellular region for laser dissection? If medial region is the major region for FC tension, how does this tension change by BM stiffness transfer inside to balance the NC tension? Can authors comment on this unclear point?

Minor comments:

1. Authors are still not careful of the stage of egg chamber and BC migration for the comparison of different genotypes, such as Supple Movie 2, Supple Fig 3, Supple Fig 9. Different migratory stages can strongly affect the migration speed for sample comparison.

2. Authors argued that less tension in NCs can prevent BC migratory rotation while strong NC tension favors BC rotation. However, we noticed that in Supple movie 9 (Abi RNAi expression in NCs), BC rotation mode is even stronger than control. For rotation experiment, can authors use longer time period for this behaviour, since short-time movies (used by authors for BM stiffness changes, compared with control) might miss the rotation occurrence in some cases.

3. Authors mentioned twice in the text that protrusions of BCs are filopodia. However, without prominent markers of filopodia, it is not precise to judge whether protrusions are filopodia only based on their finger-liked shape. One known example is Cadherin fingers in endothelial cells which are not filopodia structure at all.

4. Line201-205 "Quantification of BC RNAi migration speed from live imaging analysis revealed that BCs from tslGFP; tj>LanB1RNAi (n=7), tslGFP; LanB1hyp (n=5) or tslGFP; tj>Coll I (n=5) egg chambers moved faster than BCs from control follicles (tslGFP; tj>Gal4, n=8) (Fig.1A-ED-H, Movie S2S1, see Materials and Methods)".

Should be "Quantification of BC RNAi migration speed from live imaging analysis revealed that BCs from tslGFP; tj>LanB1RNAi (n=7), tslGFP; LanB1hyp (n=5) and tslGFP; tj>vkgRNAi (n=5) egg chambers moved faster than BCs from control follicles (tslGFP; tj>Gal4, n=8) (Fig.1A-ED-H, Movie S2S1, see Materials and Methods)".

5. None of Supple movies had a scale bar, and Supple Movie 2/7 missed the time points.

---

## [Editor Report · Decision Letter 2]

24 May 2023

Dear Dr Martin-Bermudo,

My name is Luke Smith - I am the cover editor assigned to your PLOS Biology manuscript "Constriction Imposed by Basement Membrane RegulatesDevelopmental Cell Migration" which I am handling this week, on behalf of my colleague, Richard Hodge, who is out of the office at the moment. Thank you for your patience while we considered your revised manuscript. This revised version of your manuscript has been evaluated by the PLOS Biology editors and the Academic Editor, Anna Kicheva. Overall, we are fully satisfied by your response to reviewers and to the changes made in response to our previous editorial requests. Therefore, on behalf of my colleagues and the Academic Editor, I am pleased to say that we can in principle accept your manuscript for publication, provided you address any remaining formatting and reporting issues. These will be detailed in an email you should receive within 2-3 business days from our colleagues in the journal operations team; no action is required from you until then. Please note that we will not be able to formally accept your manuscript and schedule it for publication until you have completed any requested changes.

****We did notice a minor typo in the new version of your title - which is missing a space (RegulatesDevelopmental). As you address the formatting and reporting requests, to come, please make sure to fix that as well. 

PRESS

Sincerely, 

Luke Smith, PhD

Associate Editor

PLOS Biology 

lsmith@plos.org

--on behalf of--

Richard Hodge, PhD

Associate Editor

PLOS Biology

rhodge@plos.org